# High N-glycan multiplicity is critical for neuronal adhesion and sensitizes the developing cerebellum to N-glycosylation defect

Daniel Medina-Cano[1,2], Ekin Ucuncu[1,2], Lam Son Nguyen[1,2], Michael Nicouleau[1,2], Joanna Lipecka[3], Jean-Charles Bizot[4], Christian Thiel[5], François Foulquier[6], Nathalie Lefort[7], Catherine Faivre-Sarrailh[8], Laurence Colleaux[1,2], Ida Chiara Guerrera[3], Vincent Cantagrel[1,2]*

[1]Paris Descartes-Sorbonne Paris Cite´ University, Paris, France; [2]Developmental Brain Disorders Laboratory, Imagine Institute, INSERM UMR 1163, Paris, France; [3]Proteomics platform 3P5-Necker, Université Paris Descartes - Structure Fédérative de Recherche Necker, INSERM US24/CNRS UMS3633, Paris, France; [4]Key-Obs SAS, Orléans, France; [5]Center for Child and Adolescent Medicine, Kinderheilkunde I, University of Heidelberg, Heidelberg, Germany; [6]Université Lille, UMR 8576 – UGSF - Unité de Glycobiologie Structurale et Fonctionnelle, CNRS, Lille, France; [7]iPS Core Facility, Imagine Institute, Paris, France; [8]Aix Marseille Université, INSERM UMR1249, Marseille, France

**Abstract** Proper brain development relies highly on protein N-glycosylation to sustain neuronal migration, axon guidance and synaptic physiology. Impairing the N-glycosylation pathway at early steps produces broad neurological symptoms identified in congenital disorders of glycosylation. However, little is known about the molecular mechanisms underlying these defects. We generated a cerebellum specific knockout mouse for *Srd5a3*, a gene involved in the initiation of N-glycosylation. In addition to motor coordination defects and abnormal granule cell development, *Srd5a3* deletion causes mild N-glycosylation impairment without significantly altering ER homeostasis. Using proteomic approaches, we identified that *Srd5a3* loss affects a subset of glycoproteins with high N-glycans multiplicity per protein and decreased protein abundance or N-glycosylation level. As IgSF-CAM adhesion proteins are critical for neuron adhesion and highly N-glycosylated, we observed impaired IgSF-CAM-mediated neurite outgrowth and axon guidance in *Srd5a3* mutant cerebellum. Our results link high N-glycan multiplicity to fine-tuned neural cell adhesion during mammalian brain development.
DOI: https://doi.org/10.7554/eLife.38309.001

*For correspondence:
vincent.cantagrel@inserm.fr

Competing interests: The authors declare that no competing interests exist.

## Introduction

Protein N-glycosylation, one of the most abundant post-translational modification, helps direct various cellular functions, such as protein folding, stability, trafficking and localization(*Cherepanova and Gilmore, 2016*) (*Freeze et al., 2014*). Nearly all proteins transported through the secretory pathway undergo N-glycosylation, particularly to regulate cell surface abundance and cellular interactions (*Dennis et al., 2009*). This template-independent process shows distinct glycosylation patterns that vary by protein and physiological context(*Dennis et al., 2009*). Protein N-glycosylation begins in the endoplasmic reticulum (ER), where a tightly controlled and conserved biosynthetic pathway

synthesizes a precursor named the lipid-linked oligosaccharide (LLO). This LLO comprises a sequential assembly of 14 monosaccharides ($Glc_3Man_9GlcNAc_2$) on top of the phosphorylated lipid carrier dolichol. Then, the oligosaccharyltransferase (OST) complex transfers *en bloc* the oligosaccharide part of the LLO to Asn residues on specific sites (Asn-X-Ser/Thr, X≠Pro) in nascent glycoproteins (*Chavan and Lennarz, 2006*). N-linked glycan undergoes final modifications in the ER and Golgi compartments.

Disrupting the N-glycosylation process in humans causes congenital disorders of glycosylation (CDG), a wide and highly heterogeneous group of inherited metabolic disorders (*Ng and Freeze, 2018*) (*Freeze et al., 2014*) (*Jaeken and Péanne, 2017*). The analysis of the clinical and metabolic consequences of each defect is challenging as the underlying mutated enzymes or transporters are often involved in multiple and intricate pathways(*Freeze et al., 2014*). Among them, disrupted synthesis or transfer of LLO underlies the most prevalent disorder, CDG type I (CDG-I)(*Jaeken and Péanne, 2017*). CDG-I diagnosis is based on the detection of hypoglycosylation of the patients' serum transferrin that exhibits unoccupied N-glycosylation site(s). Clinically, CDG-I disorders present psychomotor retardation (the most common feature) associated with cerebellar ataxia, seizures, retinopathy, liver fibrosis, coagulopathies, abnormal fat distribution and failure to thrive. Cerebellar ataxia is the primary diagnostic indicator in patients with *PMM2* mutations (PMM2-CDG, also known as CDG-Ia disorder), which is the most frequent CDG (*Schiff et al., 2017*). These phenotypic defects may arise from unoccupied glycosylation sites (i.e. protein hypoglycosylation) in numerous and mostly unidentified proteins.

The efficacy of protein glycosylation relies on the primary sequence at the glycosylation site (i.e. the sequon), the neighboring amino acids and the local structure (*Poljak et al., 2018*) (*Murray et al., 2015*). In CDG, the limited amount of LLO, the OST complex substrate, can also impact glycosylation site occupancy to cause hypoglycosylation (*Burda and Aebi, 1999*) (*Freeze et al., 2015b*). Initial studies investigated the consequences of this metabolic defect mainly in serum (*Hülsmeier et al., 2007*) (*Richard et al., 2009*). However, tissue accessibility and technical limitations precluded examining defects in organs, specifically the developing brain (*Freeze et al., 2015b*). The lack of data related to the consequence of the defect at the proteomic level contrasts with the well-known importance of N-glycans in specific phases of neural development (*Scott and Panin, 2014b*).

While several CDG models showing neural developmental defects exist, the mis-glycosylated targets remain unidentified (*Chan et al., 2016*; *Cline et al., 2012*; *Scott and Panin, 2014a*). We still do not know whether some specific proteins show a predisposition for these alterations, which contribute to disease pathogenesis. We predict that some glycoproteins possess a proclivity for hypoglycosylation based on their intrinsic properties (*Poljak et al., 2018*) (*Hülsmeier et al., 2007*). Previous studies identified SRD5A3, an enzyme involved in LLO synthesis, as mutated in CDG type I patients (SRD5A3-CDG) (*Cantagrel et al., 2010*) (*Morava et al., 2010*). SRD5A3 is a steroid-reductase-like enzyme involved in the last step of de novo synthesis of dolichol, the lipid used to build the LLO precursor. The broad clinical spectrum observed in patients with *SRD5A3* mutations shows many similarities with other ER-related glycosylation defects including psychomotor retardation and cerebellar ataxia. These symptoms result from likely null *SRD5A3* alleles (*Kara et al., 2014*; *Tuysuz et al., 2016*) (*Wheeler et al., 2016*) (*Cantagrel et al., 2010*) (*Gründahl et al., 2012*) and reflect impaired protein N-glycosylation, as previously described in yeast and mouse embryos mutated for the corresponding *SRD5A3* ortholog (*Cantagrel et al., 2010*). Here, we sought to gain mechanistic and functional insight into protein N-glycosylation during neural development, so we used conditional disruption of the *Srd5a3* gene in the mouse cerebellum, a tissue often affected in CDG.

## Results

### Conditional deletion of *Srd5a3* in the whole developing cerebellum

Since we sought to examine the function of *SRD5A3* in brain disease pathogenesis, we generated targeted conditional and null alleles of mouse *Srd5a3*, flanking exons 2 – 4 with loxP sequences (*Figure 1—figure supplement 1A,B*). Homozygous germline mutants (*Srd5a3*$^{-/-}$) are embryonic lethal (data not shown) consistent with results from the *Srd5a3* gene-trap mutant (*Cantagrel et al., 2010*). So, we used the Engrailed1-cre (En1-Cre) transgenic line to produce conditional knockouts En1-Cre;

*Srd5a3*[fl/-] in the developing hindbrain (*Sgaier et al., 2007*) and confirmed the gene deletion at the transcript level (*Figure 1—figure supplement 1C–E*). En1-Cre; *Srd5a3*[fl/-] mice were fertile and showed nearly Mendelian ratios at weaning age (data not shown). We used far-western blotting (far-WB) with biotinylated Sambucus Nigra lectin (SNA) to investigate the abundance of complex sialy-lated N-glycans (*Cao et al., 2013*). Total protein extracts from mutant cerebellums showed a non-significant 12% decrease in normalized signal intensity (*Figure 1A,B*). We obtained similar results with Concanavalin A (ConA) lectin that binds to core and immature mannosylated N-glycans (*Figure 1—figure supplement 1F,G*). These results indicate that cerebellar *Srd5a3* deletion does not cause a severe general glycosylation defect. However, Lamp1, a broadly used marker for N-glycosylation defects (*Rujano et al., 2017*) (*Kretzer et al., 2016*), indicated a severe decrease in protein levels across different tissues (*Figure 1C–E*). The remaining Lamp1 showed a shift toward a lower molecular weight before PNGase treatment, which indicated impaired glycosylation (*Figure 1D,F*).

Next, we investigated the functional consequences of this cerebellar glycosylation defect on learning ability and motor behavior (*Koziol et al., 2014*) in a cohort of control and mutant mice (n = 30). In the Morris Water Maze (MWM), we exposed mice twice to the same hidden platform to test working memory. We found no significant difference on swimming speed between En1-Cre;

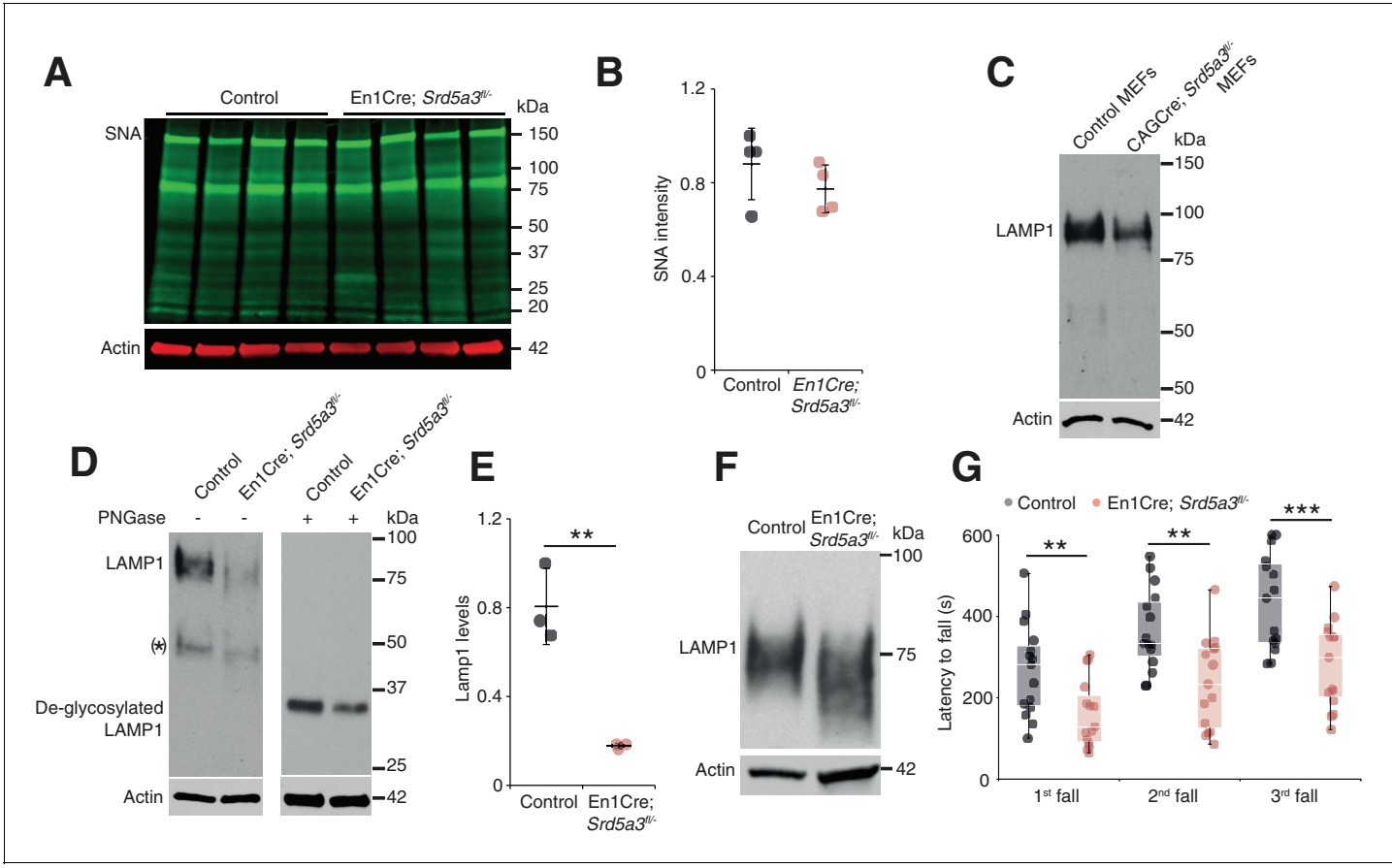

**Figure 1.** Deletion of *Srd5a3* in the cerebellum impairs protein N-glycosylation and motor function. (**A**) Far-WB with SNA lectin in P7 cerebellum and (**B**) quantification (n = 4 per genotype), p-value=0.29 (**C**) WB analysis of LAMP1 expression in mouse embryonic fibroblasts (MEFs). (**D**) WB analysis of LAMP1 level from P7 cerebellum and quantification (**E**). * indicates a PNGase sensitive LAMP1 isoform; **p-value=0.0032. (**F**) WB analysis as described in (**D**) but with increased electrophoretic migration and adjusted protein amounts to highlight LAMP1 hypoglycosylation in the mutant sample. (**G**) Box plot of the latency to fall during rotarod testing (n = 15 for each condition). One-way ANOVA was used for rotarod statistics. For all others, two-tailed student t-test was used. **p<0.01; ***p<0.001. Results are presented as mean ±s.d.

DOI: https://doi.org/10.7554/eLife.38309.002

The following figure supplement is available for figure 1:

**Figure supplement 1.** Generation of conditional *Srd5a3* knockout mice and extended biochemical and behavioral analyses.

DOI: https://doi.org/10.7554/eLife.38309.003

*Srd5a3*$^{fl/-}$ and control mice (*Figure 1—figure supplement 1H*). The latency to reach the platform was significantly lower on the second trial than on the first one for control mice, but not for En1-Cre; *Srd5a3*$^{fl/-}$ mice (*Figure 1—figure supplement 1I*). This result suggests a mild impairment of working memory in the mutant mice. In contrast, En1-Cre; *Srd5a3*$^{fl/-}$ mice showed a severe and highly significant (p=0.0007) motor coordination defect assessed with the rotarod test (*Figure 1G*). These results suggest that in the absence of *Srd5a3,* the impaired cerebellum function arises from a mild and potentially selective hypoglycosylation of glycoproteins.

## Conditional deletion of *Srd5a3* disrupts cerebellum granule cell development

Then, we wanted to examine cerebellar development after deletion of *Srd5a*3 by conducting a morphological and histological survey at P14, P21, 2 and 6 months. We observed a significant hypoplasia in the En1-Cre; *Srd5a3*$^{fl/-}$ mice (*Figure 2A–D*). Examining the mutant cerebellum cytoarchitecture revealed an accumulation of scattered ectopic cell clusters in the external part of the molecular layer (ML; *Figure 2E*). These clusters were NeuN-positive, post-mitotic granule cells (GCs) (*Figure 2F*) that failed to migrate to the internal granule cell layer (IGL). These cells remained in the external granule cell layer (EGL), a transient germinal zone. We observed this pattern at all our investigated latter timepoints with a higher incidence of ectopic clusters in the posterior lobules (*Figure 2—figure supplement 1*). Our examination of the two other major cerebellar cell populations, Purkinje cells (PCs) and Bergman glia (BG), did not reveal major cellular defects (*Figure 2—figure supplement 2A–C*). Cerebellar development relies on key glycosylated proteins expressed and secreted by PC or BG cells, such as Shh (*Dahmane and Ruiz i Altaba, 1999*) or dystroglycan (*Nguyen et al., 2013*). So, we investigated the origin of the identified GCs ectopia. We deleted *Srd5a3* using a GC-specific Cre line (Atoh1-Cre)(*Matei et al., 2005*) (*Figure 2—figure supplement 2D*). Atoh1-Cre; *Srd5a3*$^{fl/-}$ mice exhibit similar GC ectopias although in a lesser extent compared to the En1-Cre; *Srd5a3*$^{fl/-}$ mice (*Figure 2G*, *Figure 2—figure supplement 1C,D*). This observation indicates a partial GC-autonomous defect and could be explained by the involvement of GC glycoproteins and non-GC interactors that interplay to allow GCs to initiate their migration and prevent the formation of these ectopic clusters.

Taken together, our results support a critical role for *Srd5a3* during cerebellar granule cells development likely through a GC-autonomous mechanism involving the glycosylation of specific, but undetermined, proteins.

## Cerebellum-specific *Srd5a3* deletion decreases the levels of highly glycosylated proteins

Next, we evaluated the molecular mechanisms underlying our observed cerebellar defect by conducting a total proteomic analysis on the developing P7 mouse cerebellum (*Lipecka et al., 2016*; *Wiśniewski et al., 2009*). This approach quantified 1982 proteins, whose abundance profiles can cluster each sample by genotype (*Figure 3—figure supplement 1A*, *Supplementary file 1*). Our statistical analysis identified 97 differentially abundant proteins (DAP) ($\approx$ 5% of the total; q-value <0,05) in the En1-Cre; *Srd5a3*$^{fl/-}$ mice (*Figure 3A*). To determine the deregulated pathways among the highly active, neural development pathways at this stage, we performed an over-representation analysis of the 97 DAP using ConsensusPathDB. Our analysis indicated that different pathways involved in ion and amino acid transport, synapse function, cell adhesion-related signaling and cholesterol biosynthesis interplay to generate the mouse phenotype (*Figure 3B*). Aside from changes in cholesterol metabolism, all enriched pathways contained at least one deregulated N-glycoprotein. Protein N-glycosylation is critical for ER-protein folding, however we did not find any enrichment in the ER stress pathway. Manual inspection of the 97 DAP revealed increased levels of only two ER-stress-related proteins, SDF2L1 (*Fujimori et al., 2017*) and HYOU1 (*Zhao et al., 2010*), while BiP, the classical ER-stress marker(*Ron and Walter, 2007*), showed a 1.3 fold non-significant (q = 0.08) increase (*Supplementary file 1*). We then performed a transcriptomic study on additional samples at the same stage, which confirmed the absence of any significant deregulation of the most widely used UPR markers (*Figure 3—figure supplement 1B*). In addition, we did not observe any change in BIP levels restricted to ectopic GC clusters (*Figure 3—figure supplement 1C,D*). Surprisingly, our results exclude a main effect of deregulated N-glycosylation processes on ER stress.

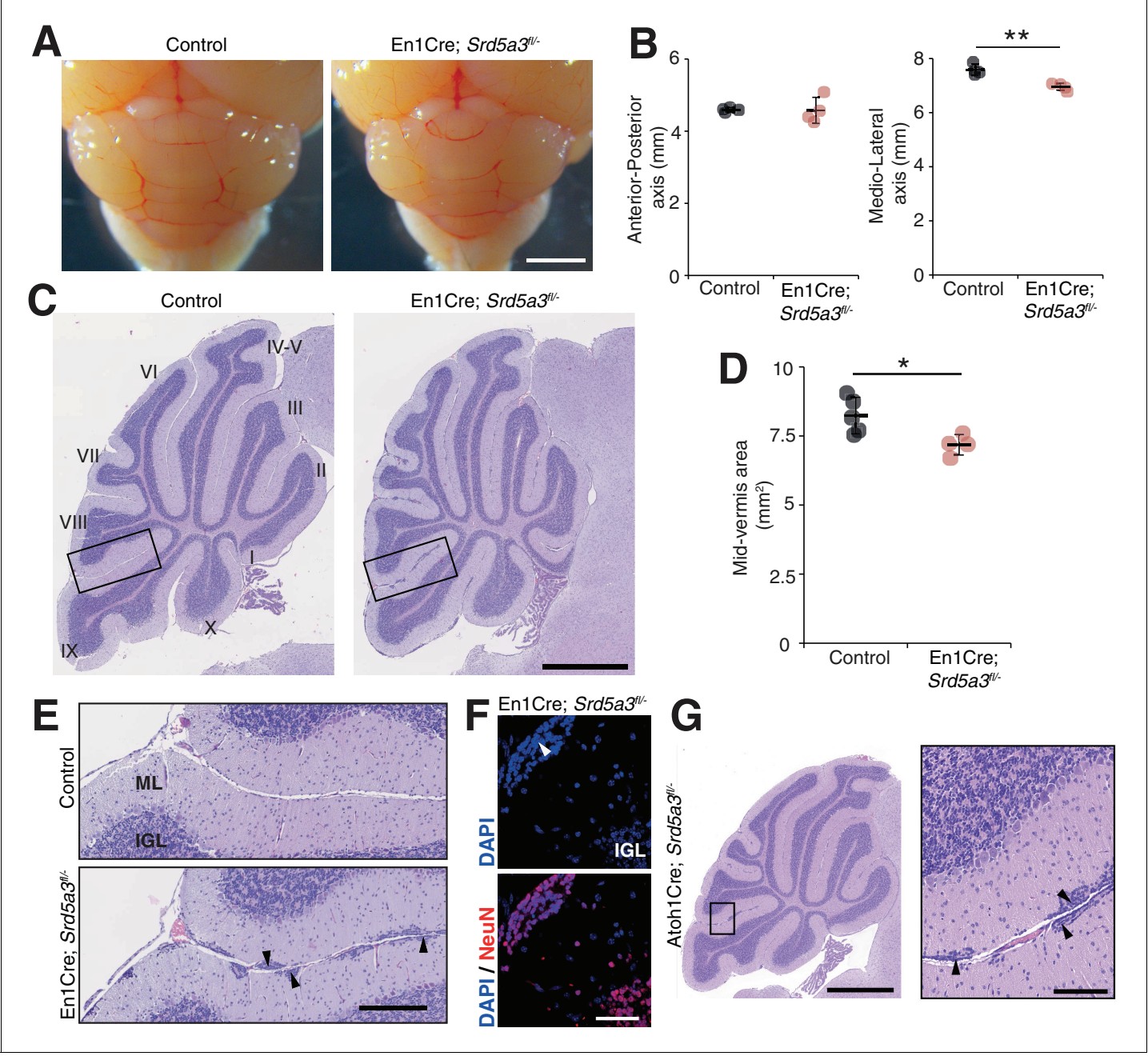

**Figure 2.** Reduced cerebellum size and ectopic granule neurons in the *Srd5a3* mutant. (A) Dorsal view of P21 posterior brains, scale bar 2 mm. (B) Quantification of the cerebellar anterior-posterior and medio-lateral axes (n = 4 for each genotype), **p-value=0.0018. (C) Representative images of hematoxylin-eosin (HE) stained P21 sagittal slices of control and mutant cerebella, scale bar 1 mm. Lobule numbers are indicated. (D) Quantification of the midline sagittal cerebellar area (Control, n = 5; En1-Cre; *Srd5a3*^*fl/-*^, n = 4); *p-value=0.0245. (E) Magnification of HE-stained cerebellar cortex, scale bar 200 μm. All examined mutants show ectopic cell clusters (arrow-head) in the outer part of the molecular layer (ML). (F) DAPI (blue) and NeuN (red) staining of P21 En1-Cre; *Srd5a3*^*fl/-*^ cerebellum. Ectopic cells (arrow-head) are positive for the post-mitotic GCs marker NeuN, scale bar 50 μm. (G) Representative image of an HE-stained sagittal slice of Atoh1-Cre; *Srd5a3*^*fl/-*^ cerebellum showing similar ectopic cells in the outer ML under GC-specific *Srd5a3* deletion, scale bar 1 mm and 100 μm, respectively (n = 4). Two-tailed Student t-test was used for statistics. *p<0.05; **p<0.01. Results are presented as mean ± s.d.

DOI: https://doi.org/10.7554/eLife.38309.004

The following figure supplements are available for figure 2:

**Figure supplement 1.** Cerebellar granule cell ectopias quantification.

DOI: https://doi.org/10.7554/eLife.38309.005

*Figure 2 continued on next page*

*Figure 2 continued*

**Figure supplement 2.** Histological analysis of Purkinje cells (PCs) and Bergmann glia (BG).

DOI: https://doi.org/10.7554/eLife.38309.006

One possible consequence of defective protein N-glycosylation is decreased stability of hypogly-cosylated proteins. We observed a significant, 4-fold enrichment of N-glycoproteins among the DAP with decreased levels (simplified as decreased N-glycoproteins; p=0.0012, *Figure 3A*). Examining this group of decreased N-glycoproteins revealed that they have more N-glycosylation sites (i.e. higher site multiplicity) with an average of 5.57 sites versus 3.24 in unchanged N-glycoproteins

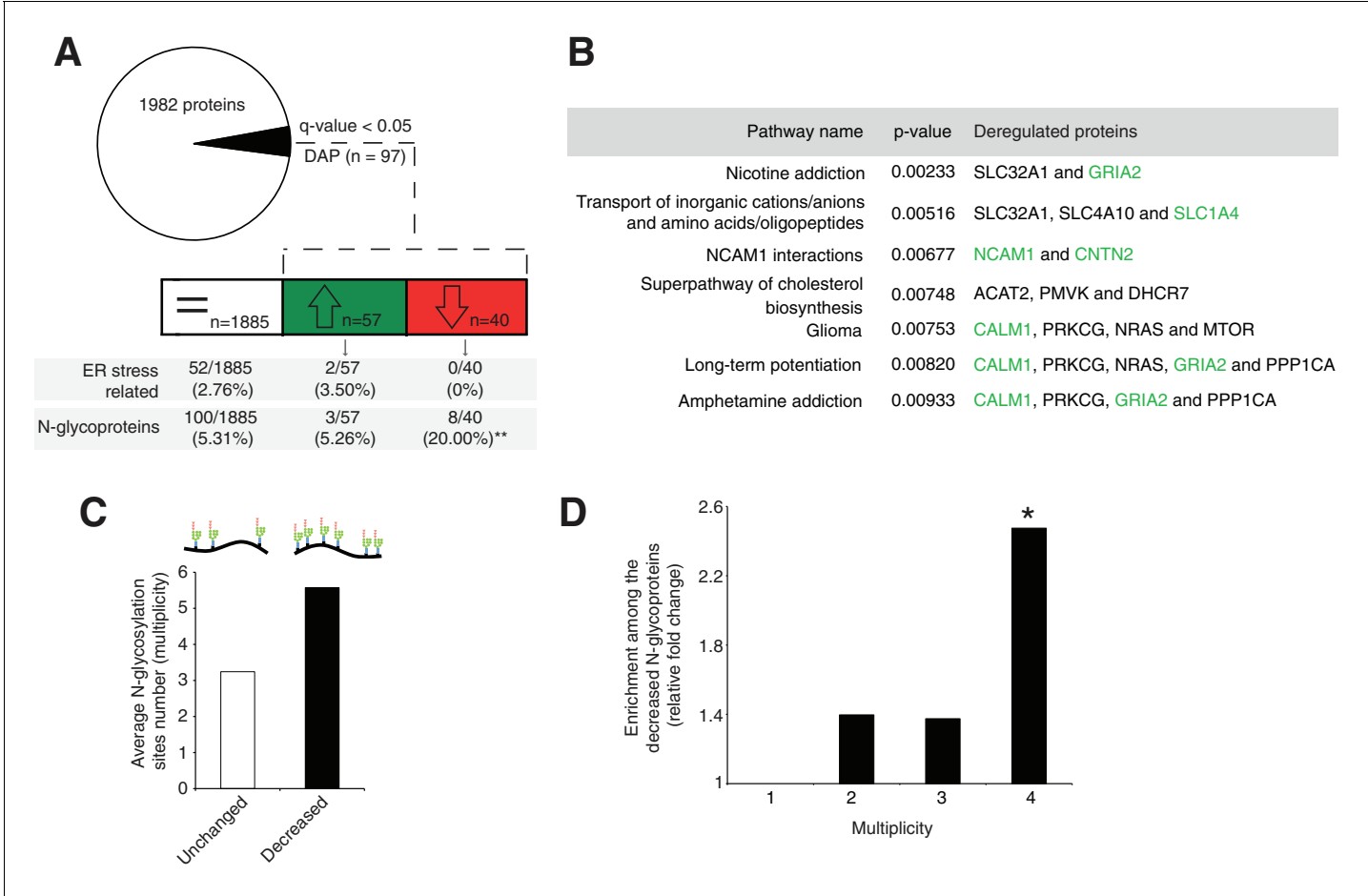

**Figure 3.** Disrupted neural pathways in the *Srd5a3* mutant developing cerebellum. (**A**, upper panel) Pie chart representing the total proteomics analysis in P7 cerebellum (n = 4 for each genotype). 97 proteins were detected as differentially abundant proteins (DAP) in the mutant samples (q-value <0.05). (**A**, lower panel) Distribution of unchanged proteins (white box), increased DAP (green box) and decreased DAP (red box) into ER stress-related (GO:0034976) or N-glycoproteins (Reference glycoproteomic database, see Materials andmethod) categories. A significant enrichment was detected within the decreased DAP for N-glycoproteins (n = 8, Fisher exact test, p-value=0.0012). (**B**) Over-representation analysis on DAP performed with ConsensusPathDB using the 1982 detected proteins as background. All pathways with p-value<0.01 are represented. N-glycoproteins are indicated in green. (**C**) Average multiplicity in the unchanged and the decreased N-glycoproteins groups. (**D**) Enrichment for different glycoprotein categories among the decreased N-glycoproteins. Multiplicity represents groups of proteins with more than one (≥2), two (≥3) or three (≥4) N-glycosylation sites. Multiplicity of ≥1 represents all of the decreased N-glycoproteins and is set to one (no enrichment). There is a significant 2.4 fold-change enrichment for glycoproteins with four or more N-glycans among the decreased N-glycoproteins (Fischer exact test, p-value=0.0378).

DOI: https://doi.org/10.7554/eLife.38309.007

The following figure supplement is available for figure 3:

**Figure supplement 1.** Extended total proteomic analysis of the unfolded protein response (UPR).

DOI: https://doi.org/10.7554/eLife.38309.008

(*Figure 3C*). Indeed, N-glycoproteins with four or more N-glycosylation sites are significantly enriched within the decreased N-glycoproteins (*Figure 3D*).

These data demonstrate that the mild glycosylation defect detected in the *Srd5a3* mutant cerebellum disrupts the levels of highly glycosylated proteins, but does not disrupt ER homeostasis.

## N-glycosylation site multiplicity and primary sequence underlies the selective protein glycosylation defect

To further elucidate the N-glycosylation deficiency, we included a lectin-affinity based (ConA, WGA, RCA$_{120}$) enrichment step at the peptide level, followed by deglycosylation (*Zielinska et al., 2010*), prior proteomic analysis (*Figure 4—figure supplement 1A*, data are available via ProteomeXchange with identifier PXD009906). This enrichment can identify N-glycosylation sites and quantify the abundance of each site when they are glycosylated. Using this dataset, we identified 140 likely new glycosylation sites ($\approx$8% of the total, see Materials and method, *Supplementary file 2*) with high recurrence of non-canonical sequons ($\approx$40% of the new sites, *Supplementary file 2*). Most proteins carrying these sites are intrinsic to membrane as observed for proteins with previously identified glycosylation sites (Gene Ontology analysis in *Supplementary file 2*). In total, we identified 1404 glycopeptides detected in at least 3 out of 4 control samples (*Figure 4A*, *Supplementary file 2*). As we observed for the total proteome, we could cluster the samples according to the genotype based on the expression levels of all the glycopeptides (see Materials and method, *Figure 4—figure supplement 1B*). Total proteomics and glycoproteomic analysis yielded complementary information. Total proteomics provides information of the protein abundance level but can only detect peptides containing an unoccupied N-glycosylation site. Glycoproteomics only allows detection of glycopeptides if the site is occupied. For 15% of the N-glycopeptides detected in glycoproteomics (n = 206), we found decreased levels in the mutant samples. Of these, we only detected 24 N-glycopeptides in the previous total proteomic experimental settings, consistent with the high enrichment observed with the glycoproteomics analysis (*Figure 4—figure supplement 1C*, *Supplementary file 3*). Among these peptides, 13 were never detected in control but only in mutant samples in total proteomics. These data indicate that these 13 peptides were not glycosylated in the mutant. We conclude that most differences we detected in the glycoproteomic experiment reflect reduced glycosylation site occupancy.

We also detected an extremely significant correlation between N-glycan multiplicity and N-glycopeptide levels (*Figure 4B*, Mann-Whitney test, p-value<0.0001). This result is consistent with our larger glycoprotein dataset. To delineate the origin of reduced occupancy of individual N-glycosylation sites, we analyzed their primary sequences (*Figure 4C*). We identified a mild effect of the presence of a non-aromatic amino-acid (other than Phe, Tyr, His or Trp) at position −2 and a more significant effect of the Serine at position +2 (*Figure 4D*), as previously described(*Gavel and Heijne, 1990*; *Murray et al., 2015*), with an increased impact in highly glycosylated proteins (*Figure 4E*). These findings demonstrate that N-glycan multiplicity combined with sequon efficiency is a major parameter driving protein sensitivity to N-glycosylation defects.

## Impaired protein N-glycosylation targets IgSF-CAMs

We next wanted to delineate the affected pathways based on our glycoproteomic results. We performed an over-representation analysis using proteins carrying at least one N-glycopeptide significantly decreased in mutant samples (145 N-glycoproteins, 206 N-glycopeptides). This analysis revealed that the most significantly affected proteins (4 out of the top 10 deregulated pathways) derive directly from cell adhesion and axon guidance-related pathways involving adhesion proteins from the L1CAM family (*Figure 5A*, red circles). L1CAMs and the previously identified NCAM1 and CNTN2 (*Figure 3B*) belong to the immunoglobulin superfamily of cell adhesion molecules (IgSF-CAMs) with critical roles in brain development (*Maness and Schachner, 2007*; *Pollerberg et al., 2013*; *Stoeckli, 2010*). We speculate that this particular sensitivity of IgSF-CAMs to a N-glycosylation defect arises from their higher N-glycosylation site multiplicity with an average of 3.3 sites versus 2.3 in non IgSF-CAM proteins (Calculated based on the reference glycoproteomic dataset, *Supplementary file 2*, Mann Whitney test, p-value<0.0001).

So, we next hypothesized that impaired IgSF-CAMs function derived from sub-optimal N-glycosylation and that defective IgSF-CAMs contributed to our observed histological defect.

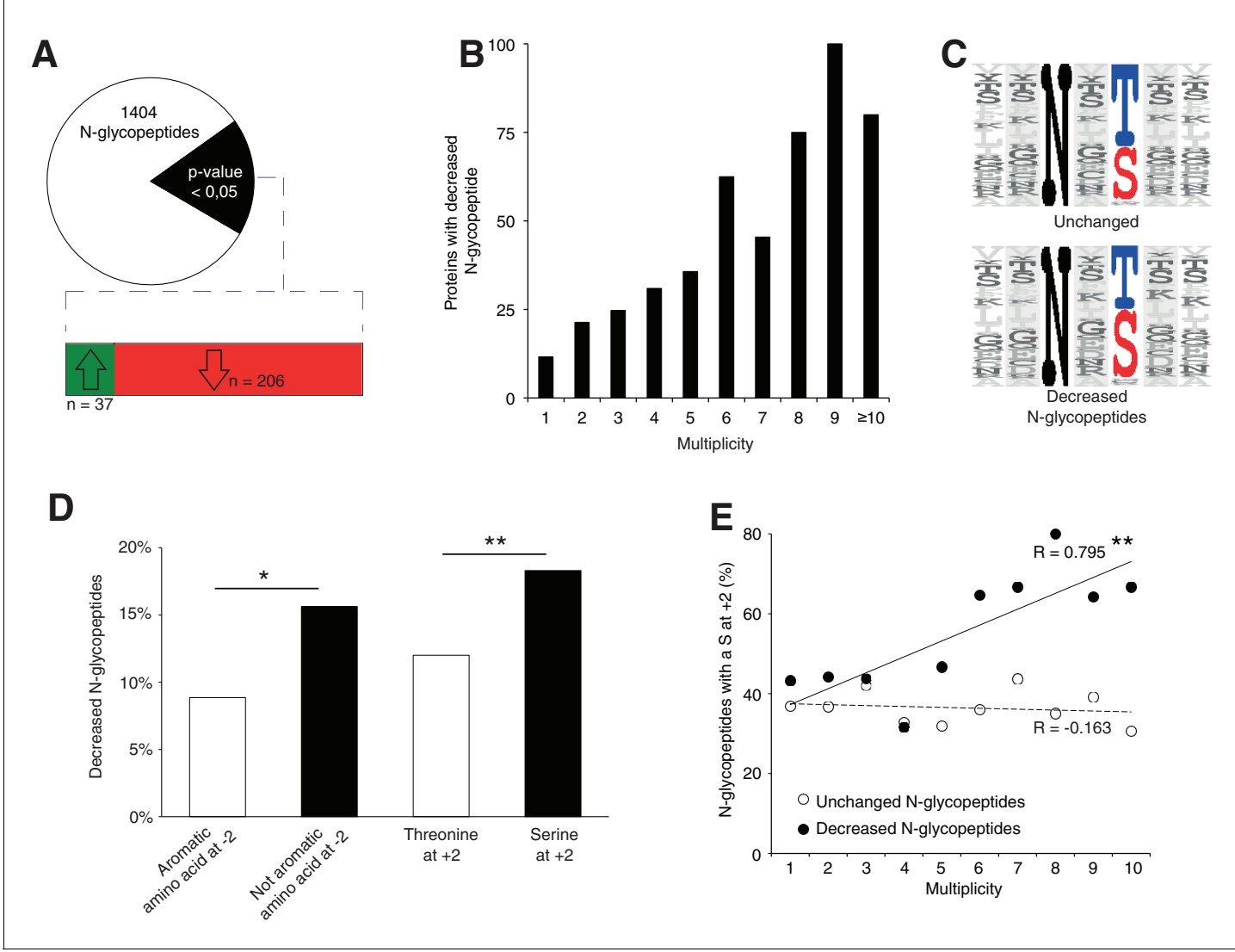

**Figure 4.** *Srd5a3* deletion impairs protein N-glycosylation in a sequon and multiplicity-dependent manner. (**A**) Pie chart representation of the glycoproteomic analysis in P7 cerebellum (n = 4 for each genotype). The vast majority of the differentially abundant N-glycopeptides was decreased in the mutant samples (206/241, 85%). (**B**) Percentage of proteins with decreased N-glycopeptide according to their multiplicity (Mann-Whitney test, p-value<0.0001). Proteins with higher multiplicity are more likely to have decreased N-glycopeptide levels. (**C**) Motif analyses of the N-glycopeptides unchanged or decreased in the mutant. (**D**) Significant enrichment within the decreased N-glycopeptides for a non-aromatic amino acid (other than Phe, Tyr, His or Trp) in position −2 and a Ser in position +2 (NxS motif). (**E**) Correlation between protein multiplicity and their percentages of decreased (black) or unchanged (dashed line) glycopeptides with NxS sequon in the mutant samples. A significant positive correlation was found between the percentage of NxS-containing glycopeptides with decreased occupancy in the mutant and the N-glycoprotein multiplicity (Pearson's coefficient, p-value=0.0062). NxS-containing N-glycopeptides are more likely to have decreased level when located in a highly glycosylated protein. Unless indicated, two-tailed Student t-test was used for statistics. *p<0.05; **p<0.01.

DOI: https://doi.org/10.7554/eLife.38309.009

The following figure supplement is available for figure 4:

**Figure supplement 1.** Extended glycoproteomic analysis results and validation.

DOI: https://doi.org/10.7554/eLife.38309.010

We confirmed the enrichment for IgSF-CAMs found in our over-representation analysis at the N-glycopeptide level (18.4% of the decreased N-glycopeptides belong to an IgSF-CAM protein, Fisher exact test, p-value=0.0058, *Figure 5B*).

We selected two highly N-glycosylated IgSF-CAMs relevant for the granule cell histological phenotype for validation by WB. We did observe a clear hypoglycosylation defect for L1CAM and

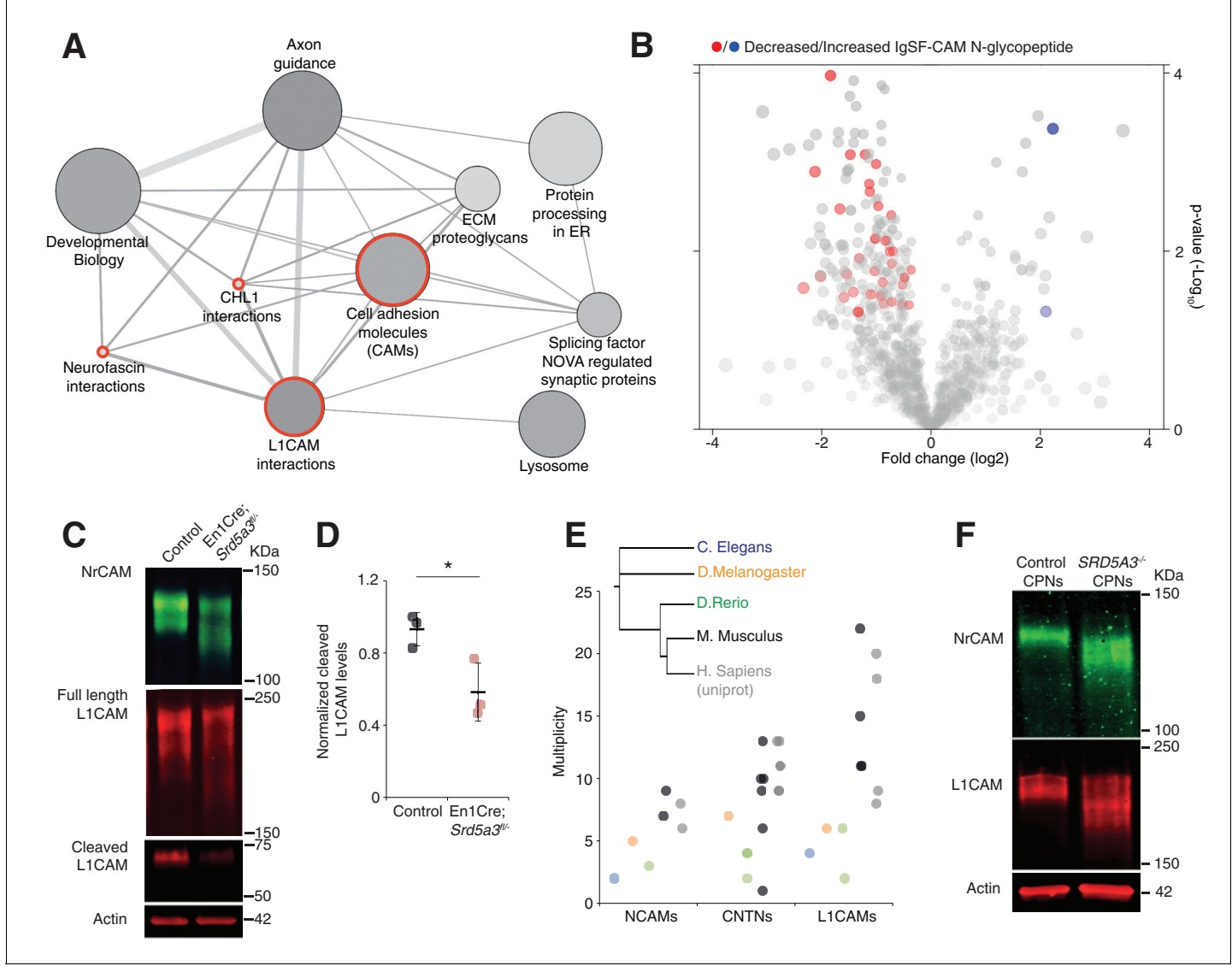

**Figure 5.** IgSF-CAMs are especially sensitive to a mild N-glycosylation impairment. (A) Graphical representation of enriched pathways and their interactions within proteins with reduced N-glycopeptide in the mutant (ConsensusPathDB). The nodes size indicates the number of proteins within the pathway, and their shade notes the significance (the darkest, the more significantly enriched). Red circles enclose the L1CAM family-related pathways. (B) Volcano plot against all N-glycopeptides. The decreased (red) or increased (blue) N-glycopeptides belonging to an IgSF-CAM protein are highlighted. There is enrichment for IgSF-CAMs glycopeptides within the decreased ones (Fisher exact test, p-value=0.0058). (C) WB against L1CAM and NrCAM in the P7 mouse cerebellum and (D) quantification of the 70 kDa cleaved isoform of L1CAM (n = 3 for each genotype, p-value=0.0312). (E) Multiplicity per protein of three IgSF-CAMs subfamilies (L1CAMs, CNTNs and NCAMs) in different species. Multiplicity information was collected from experimental datasets (*Zielinska et al., 2010*) and (*Zielinska et al., 2012*) except for human information extracted from uniprot database. (F) WB analysis of L1CAM and NrCAM levels in CPNs (n = 2 per genotype, repeated twice). Unless indicated, two-tailed Student t-test was used for statistics. Results are presented as mean ±s.d. *p<0.05.

DOI: https://doi.org/10.7554/eLife.38309.011

The following figure supplements are available for figure 5:

**Figure supplement 1.** Schematic representation of the mouse L1CAM and NrCAM with their hypoglycosylated sites.
DOI: https://doi.org/10.7554/eLife.38309.012

**Figure supplement 2.** hiPSCs differentiation towards human cortical projection neurons (CPNs).
DOI: https://doi.org/10.7554/eLife.38309.013

NrCAM with a detected shift to lower molecular weights and a significant decrease in the glycosylation-dependent cleavage of L1CAM (*Lutz et al., 2014*) (*Figure 5C,D*, *Figure 5—figure supplement 1*).

A major role for IgSF-CAMs in mammalian development is illustrated by the increased number of their encoding genes across evolution (*Hortsch, 2000*). We also observed an increased in the N-glycan multiplicity for three IgSF-CAM families with critical roles in brain development (*Figure 5E*). The multiplicity of these glycoproteins is almost systematically much higher in mammals compared to the other species. To test if the IgSF-CAM sensitivity for glycosylation impairment is conserved, we generated *SRD5A3* knockout (KO) human induced pluripotent stem cells (hiPSCs). We then differentiated them toward cortical projection neurons (CPNs, *Figure 5—figure supplement 2*). As previously observed in mouse, human L1CAM and NrCAM immuno-blotting profiles exhibit clear shifts in *SRD5A3*$^{-/-}$ CPNs (*Figure 5F*). These data demonstrated a similar exacerbated sensitivity of IgSF-CAMs to SRD5A3 loss in human neurons. Our data suggest that the sensitivity of highly glycosylated IgSF-CAMs to N-glycosylation defect is conserved across mammalian species and that the increase in multiplicity during evolution parallels the acquisition of a more complex neural cell organization.

## Hypoglycosylation of L1CAM and NrCAM alters their expression level at the plasma membrane

In order to evaluate the consequences of IgSF-CAM hypoglycosylation at the cellular level, we studied the cell surface expression of L1CAM and NrCAM using antibodies directed against their ectodomain in non-permeabilized cells (see Materials and methods). We detected the expression of these IgSF-CAMs along the axons and at the growth cone but with 30.8% and 39.6% decrease for L1CAM and NrCAM, respectively, in *Srd5a3* cKO neurons (*Figure 6A,B*). We recapitulated a similar defect using Tunicamycin suggesting that the disruption of other steps of the N-glycosylation pathway as well as specific drugs can efficiently alter the cell-surface expression of these glycoproteins. Such differences in the membrane expression levels are expected to have consequences on the ability of neurons to bind their substrates. Both L1CAM and NrCAM are known to interact homophilically and heterophilically with multiple IgSF-CAM, e. g. CNTN1 and CNTN2 (*Falk et al., 2002*; *Sonderegger, 1997*). We decided to test for the homophilic binding activity, by incubating cultured GCs with chimeras made of the extracellular part of L1CAM or NrCAM fused with the IgG Fc region (L1CAM-Fc and NrCAM-Fc, respectively). Indeed, a 38.5% and 56.0% decrease in the binding of L1CAM and NrCAM chimeras, respectively, was observed in mutant cells (*Figure 6C,D*). This result validates the functional impact of IgSF-CAMs hypoglycosylation and suggests that the glycosylation level of IgSF-CAMs plays a critical role in the regulation of their cell-surface expression.

## *Srd5a3* is necessary for IgSF-CAM-dependent cerebellar granule cells adhesion and axon guidance

IgSF-CAMs rely on their glycan charge to interact with each other (*Wei and Ryu, 2012*) (*Horstkorte et al., 1993*). Proper IgSF-CAMs trans-interaction is essential for adequate nervous system connectivity, for fasciculation and axonal guidance (*Pollerberg et al., 2013*). To demonstrate that IgSF-CAMs hypoglycosylation is directly involved in En1-Cre; *Srd5a3*$^{fl/-}$ mouse phenotype, we examined the neurite dynamics by live cell imaging of isolated GCs cultured under different surface coating conditions. *En1-Cre; Srd5a3*$^{fl/-}$ GCs did not show any significant alterations that suggested differences in proliferation or cell death onto any substrates (*Figure 7—figure supplement 1A*). In contrast, GCs showed defective neurite sprouting with a 20% decrease when using a coating with laminin and poly-D-lysine (*Figure 7A,B*). This phenotype was exacerbated in the presence of coatings made of human recombinant L1CAM or NrCAM proteins that enhance homophilic interactions (*Dequidt et al., 2007*). We observed a 37% and 52% reduction in neurite number for mutant GCs compared to control, on L1CAM and NrCAM substrates, respectively (*Figure 7C*). A significant decrease was also observed in neurite length (*Figure 7—figure supplement 1B*) but less specifically associated with IgSF-CAM coating (*Figure 7—figure supplement 1B–D*). To further link the IgSF-CAM-dependent neurite development in the mutant GCs to the defective initiation of cell migration, we used GC re-aggregates to evaluate cellular migration in vitro under different coating conditions (*Bizzoca et al., 2003*) (*Katic et al., 2014*; *Kerjan et al., 2005*). Using laminin and poly-D-lysine coating, the mutant re-aggregates showed no significant difference in neurite number and only a mild

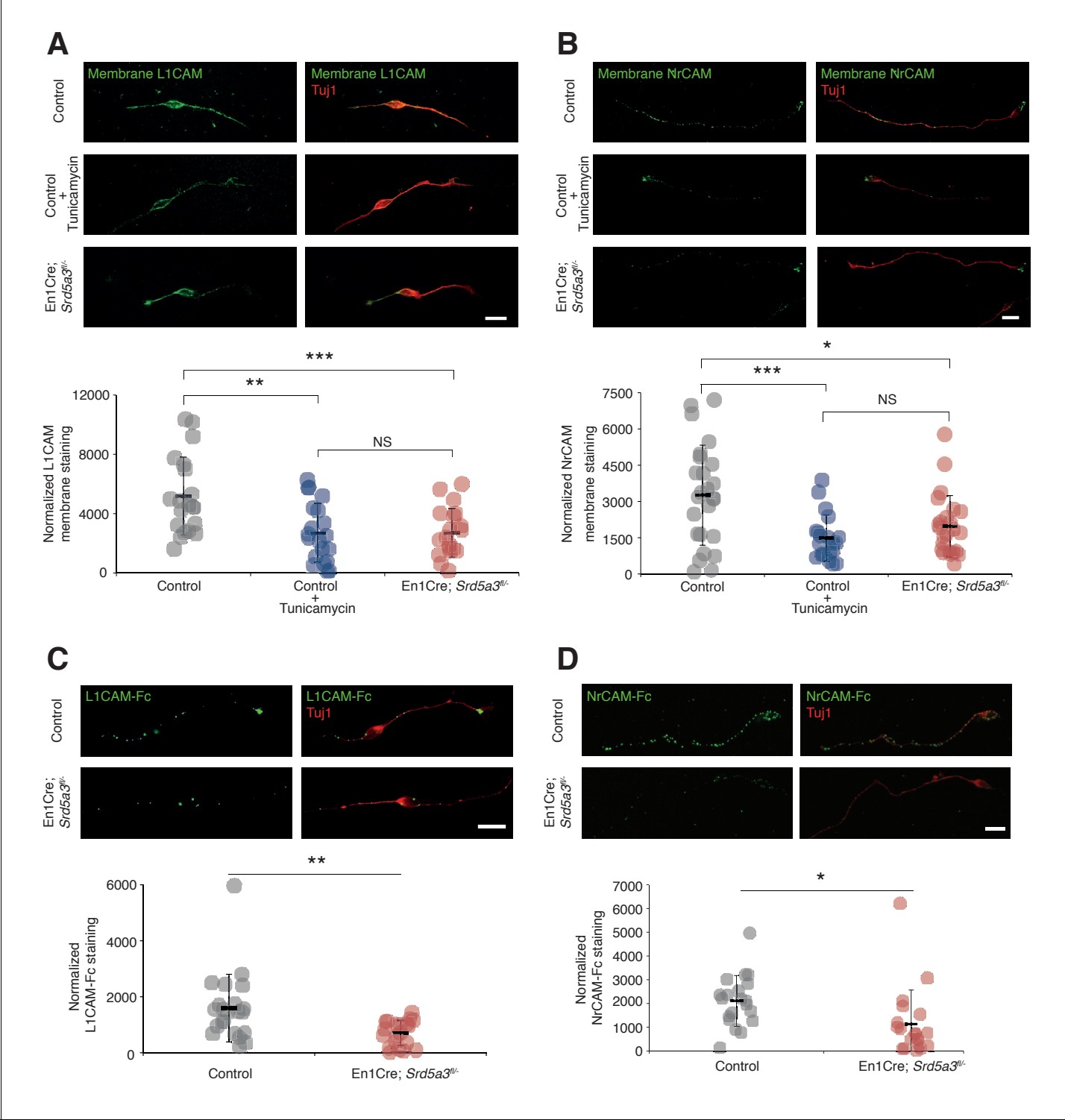

**Figure 6.** L1CAM and NrCAM hypoglycosylation affects their cell surface expression and binding activity. (**A**) L1CAM and (**B**) NrCAM surface immunostaining of cultured GCs and quantification. (**C**) L1CAM-Fc and (**D**) NrCAM-Fc binding assay on cultured GCs. The cells were incubated with the recombinant proteins and bound L1CAM-Fc or NrCAM-Fc were visualized with anti-Fc antibody. All of the staining and quantifications were performed with at least three biological replicates from different litters. Each dot represents a single neuron. Scale bar 10 µm. Results are presented as mean (black line)±s.d.

DOI: https://doi.org/10.7554/eLife.38309.014

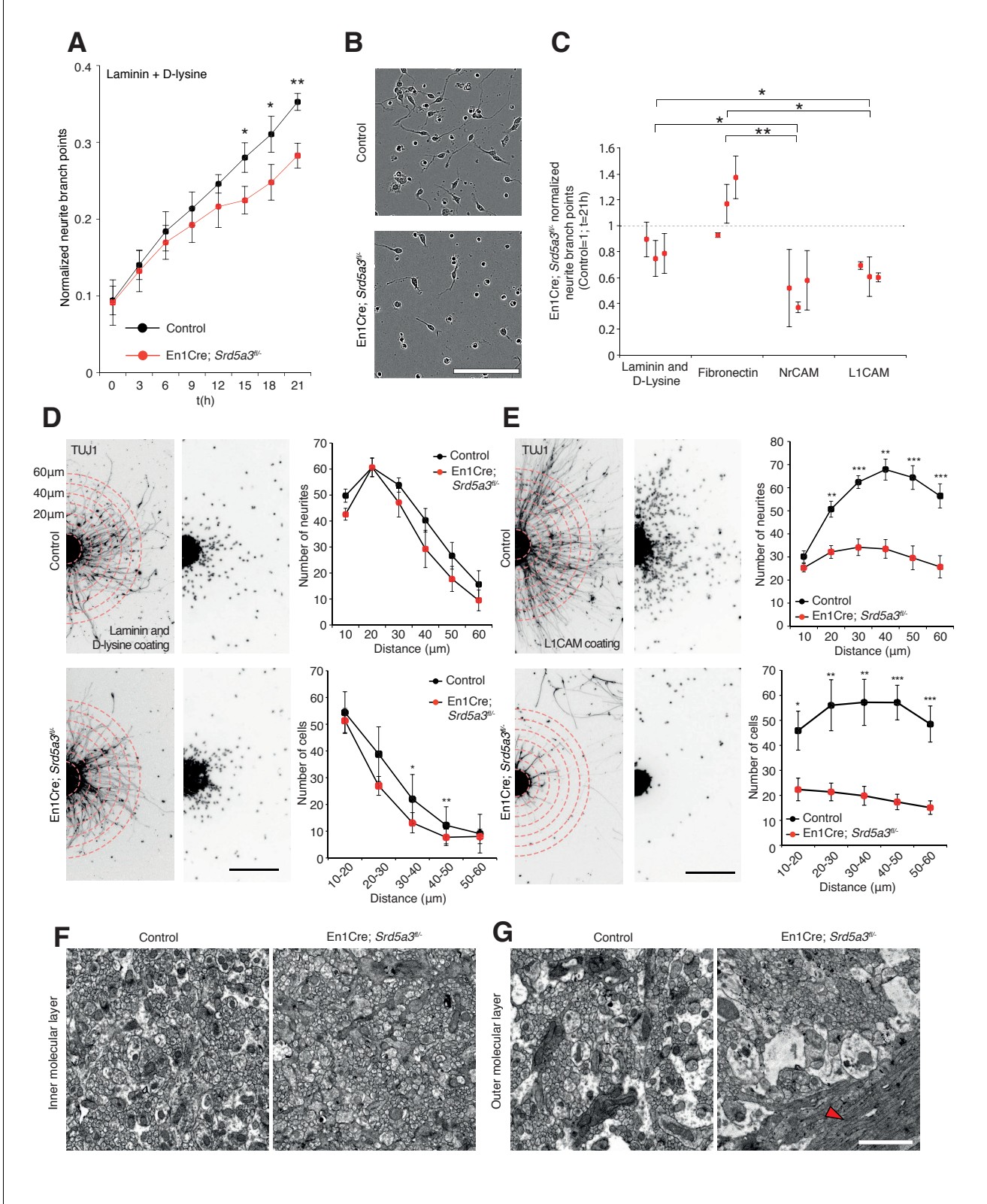

**Figure 7.** IgSF-CAM-dependent neurite dynamic and axon orientation defects in *Srd5a3* mutant cerebellum. (**A**) Neurite number per GC body cluster across 21 hr in laminin and poly-D-lysine-coated wells as measured with Incucyte live cell imaging system (Control, n = 3; En1-Cre; *Srd5a3*<sup>fl/-</sup>, n = 3). (**B**) Representative image of control and En1-Cre; *Srd5a3*<sup>fl/-</sup> GCs after 21 hr in laminin and poly-D-lysine-coated wells. Scale bar 100 μm. (**C**) Neurite number normalized to GC body clusters after 21 hr with laminin/poly-D-lysine coatings, fibronectin or IgSF-CAMs-coated wells (NrCAM and L1CAM). Each dot

*Figure 7 continued on next page*

*Figure 7 continued*

per coating represents results from a single mutant mouse. All GC cultures were performed in technical triplicates (Control, n = 3; En1-Cre; *Srd5a3*^fl/-^, n = 3). (**D**) Neurite number and number of migrating neurons in GC re-aggregates plated for 24 hr on laminin and poly-D-lysine and (**E**) L1CAM coated surfaces. Five aggregates were quantified per mouse and coating condition (Control n = 5, En1Cre; *Srd5a3*^fl/-^, n = 7). (**F**) Representative electron microscopy images of cerebellar ML sagittal view at P21 taken from control (n = 3) and *Srd5a3* mutant (En1-Cre; *Srd5a3*^fl/-^ n = 1; Atoh1-Cre; *Srd5a3*^fl/-^ n = 2) mice. Scale bar 2 μm. Parallel fibers show a single orientation in control ML, whereas some exhibit an abnormal perpendicular orientation in the most outer ML in the mutant mice (G, red arrowhead). Two-tailed Student t-test was used for statistics. *p<0.05; **p<0.01. Results are presented as mean ± s.d.

DOI: https://doi.org/10.7554/eLife.38309.015

The following figure supplement is available for figure 7:

**Figure supplement 1.** Changes in granule cell (GC) number and neurite length under different coating conditions.
DOI: https://doi.org/10.7554/eLife.38309.016

defect in migration (*Figure 7D*). However, when the aggregates were plated on L1CAM-coated surfaces, a major defect in both neurite number and subsequent cell migration was observed (*Figure 7E*). This result highlights the substrate specificity of the adhesion phenotype and overall these data show that the correct glycosylation of IgSF-CAM is critical for neurite outgrowth, especially in the initiation step. Interestingly, GCs axons development is a process CNTN2 dependent that precedes the initiation of GCs migration towards the IGL (*Xenaki et al., 2011*).

Given the prominent role of many IgSF-CAMs in axon guidance and the granule cells adhesion defect, we examined axon orientation in GCs. We analyzed the parallel fibers (PFs) organization in the cerebellar ML using electronic microscopy. In control samples, PFs are consistently oriented perpendicular to PCs branches. However, *Srd5a3* mutant mice exhibit an aberrant orientation of groups of PFs in the outer ML, likely due to defective axonal guidance (*Figure 7F,G*). Taken together, our data suggest that IgSF-CAMs hypoglycosylation may underlie many neurological defects in CDG.

## Discussion

We sought to delineate the impact of *Srd5a3*-driven hypoglycosylation of N-glycoproteins during development, a biochemical defect likely shared between multiple CDG. Here, we generated a conditional KO mouse that recapitulates some neurological symptoms induced by this ER glycosylation defect. We hypothesized that patients with *SRD5A3*-related and other types of CDG exhibit protein hypoglycosylation leading to cerebellar dysfunction. From a physiological perspective, we hypothesized that the characterization of the affected glycoproteins and glycosylation sites would identify new roles for protein N-glycosylation during central nervous system development. So, we combined total proteomic and glycoproteomic approaches to identify the molecular mechanisms underlying the observed cerebellar dysfunction and GC histological defects. Our strategy assessed the relative occupancy at each N-glycosylation site regardless of the N-glycan nature. The pleiotropic roles of glycans on development and synaptic transmission suggest that genetic defects in the synthesis of the N-glycan core structure would have multiple distinct and integrated consequences. We unexpectedly found that partial inhibition in N-glycosylation during cerebellar development has persistent consequences on a specific set of glycoproteins and pathways. Among them, we found that IgSF-CAMs-mediated cell adhesion is the most consistently affected. Our further investigations revealed impaired GC adhesion and axon growth, which support IgSF-CAM hypoglycosylation as the primary underlying defect. Our glycoproteomic and total proteomic analyses strongly suggest an intuitive mechanism where N-glycan multiplicity is a critical factor driving the molecular defects.

SRD5A3 acts at the last step of dolichol synthesis and facilitates also the synthesis of O-mannose-linked glycans, C-mannosylation, and glycophosphatidylinositol (GPI) anchored proteins. However, N-glycosylation is the most demanding cellular process with eight dolichol molecules needed for each N-linked glycan(*Cantagrel and Lefeber, 2011*). Defects in SRD5A3 alter lipid membrane composition that subsequently impairs protein N-glycosylation. Consistently with previous results using *Srd5a3* gene-trapped embryos(*Cantagrel et al., 2010*), several enzymes from the cholesterol/mevalonate pathway were found deregulated (*Figure 3B*). These observations indicate a regulatory mechanism that could alleviate these glycosylation defects by stimulating the mevalonate pathway (*Haeuptle et al., 2011*) (*Welti, 2013*). While we cannot exclude defects in other glycosylation

pathways, our results demonstrate that changes in N-glycosylated proteins and correlations between the N-glycan number and peptides abundance in both proteomic and glycoproteomic analyses strongly implicate the N-glycosylation pathway as the major metabolic target underlying the phenotype and neurological symptoms in SRD5A3-CDG. Surprisingly, we did not observe a signature of the UPR pathway as seen in other CDG models. This result does not exclude a minimal glycoprotein folding impairment that does not significantly alter the ER homeostasis. A potential mechanism of adaptation to this mild impairment is detected with the over-expression of the HYOU1 and SDF2L1 chaperones.

Our work shows that proteins with high N-glycan multiplicity, estimated to four or more sites, show an increased risk for hypoglycosylation to induce protein degradation or dysfunction. Moreover, differences in the sequon can have an additive effect to this increased risk. Although several highly-glycosylated proteins are frequently used as sensitive biomarker for N-glycosylation defects (*He et al., 2012*) (*Park et al., 2014*), we provide initial evidence that the total number of glycosites per protein is a major parameter that influences N-glycoproteins sensitivity. Our total proteomic analysis clearly indicates that decreased glycoprotein abundance is a direct consequence of incomplete glycosylation of highly glycosylated proteins. Yet, examining individual proteins, such as the IgSF-CAM member L1CAM, did not reveal significant differences in full-length isoform abundance by western blot but showed altered cleavage as well as reduced cell surface expression as a consequence of its hypoglycosylation. The cleaved L1CAM isoform (70 kDa) enhances cerebellar GC neurite outgrowth(*Lutz et al., 2014*), which indicates that hypoglycosylation of some proteins can impair their processing, trafficking and/or functioning without necessarily causing significant changes in their cellular abundance.

The number of N-glycans, in conjunction with their degree of branching, regulates cell proliferation and differentiation. Highly glycosylated growth factor receptors show increased cell-surface expression by forming galectin-mediated molecular lattices(*Lau et al., 2007*). We cannot exclude that some cell adhesion proteins utilizing a similar mechanism also show alterations in the *Srd5a3* mutant cerebellum. To our knowledge, neuronal galectin-IgSF-CAM interactions remain unreported. The pivotal role of IgSF-CAMs in cell migration, axonal guidance and synaptogenesis has been widely studied in mouse (*Sakurai et al., 2001*) (*Fransen et al., 1998*) (*Berglund et al., 1999*) (*Çolakoğlu et al., 2014*) (*Sytnyk et al., 2017*). However, we still understand very poorly how N-glycans contribute to these functions with few exceptions such as the established role for the polysialic acid modification of NCAM1 N-glycan chains(*Weinhold et al., 2005*). Interestingly, specific N-glycosylated residues, alpha2,3-linked sialic acid and Lewis(x) of L1CAM, CNTN1 and CNTN2 have been reported to mediate interaction with the glycopeptide CD24 modulating cell adhesion and neurite outgrowth of cerebellar neurons (*Lieberoth et al., 2009*).

Among proteins identified in our proteomic approaches, NrCAM and L1CAM show redundant functions in cerebellar GC development, in addition to their roles in axonal growth (*Sakurai et al., 2001*) (*Demyanenko et al., 1999*) (*Demyanenko et al., 2011*). CNTN2/TAG-1 contributes to neuronal polarization and axonal organization. Interestingly, the corresponding KO mouse shows ectopic GC clusters at the pial surface of the cerebellum (*Xenaki et al., 2011*), as observed in the *Srd5a3* mutant cerebellum. Mice lacking *Cntn1* show aberrant parallel fibers distribution and motor coordination defects (*Berglund et al., 1999*). Since neural IgSF-CAMs largely bind to each other homophilically and heterophilically (*Pollerberg et al., 2013*), we suspect that the observed neuronal adhesion defects arise from a cumulative effect of partial hypoglycosylation of multiple interacting partners (*Schwarz et al., 2009*) (*Stoeckli et al., 1997*). IgSF-CAMs show increased complexity and members in vertebrates (*Vogel and Chothia, 2006*). The increased diversity in some IgSF-CAM proteins correlates with the presence of homologs with higher multiplicity in mammals (*Figure 5E*). We speculate that a positive selection of adhesion proteins with high N-glycosylation site multiplicity can support the increased complexity of mammalian neuronal organization and connectivity.

In addition to the central nervous system, highly glycosylated IgSF-CAM members also play critical roles in other systems affected in CDG patients, such as the developing eyes (*Morava et al., 2009*) and immune system with ICAM1 (*Blank et al., 2006*; *He et al., 2014*), or in other diseases like cancer, where L1CAM plays a critical role in tumor metastasis (*Kiefel et al., 2012*) (*Agrawal et al., 2017*). Further investigations will determine how IgSF-CAM hypoglycosylation regulates or alters the clinical outcomes and responsivity to therapeutic strategies. In conclusion, our study demonstrates that suboptimal functioning of the ER-glycosylation machinery impairs primarily highly glycosylated

N-glycoproteins with mild or no effect on ER homeostasis. We propose that the prevalent neurological symptoms observed in SRD5A3-CDG patients and likely other types of CDG type I result from defective neural cell adhesion, caused by IgSF-CAM hypoglycosylation. Deciphering how multiple N-glycans can influence differential IgSF-CAM adhesion properties will address important unresolved neurobiological questions regarding neuronal migration, axonal outgrowth and synaptogenesis.

# Materials and methods

## Key resources table

| Reagent type (species) or resource | Designation | Source or reference | Identifiers | Additional information |
|---|---|---|---|---|
| Genetic reagent (M. musculus) | *ACTB:FLPe B6J* | JAX stock #005703 | ACTB:FLPe B6J | |
| Genetic reagent (M. musculus) | *Atoh1-Cre* | JAX stock #011104 | Tg(Atoh1-cre)1Bfri | |
| Genetic reagent (M. musculus) | CAG-Cre | JAX stock #004682 | B6.Cg-Tg (CAG-cre/Esr1*)5Amc/J | |
| Genetic reagent (M. musculus) | *CMV-Cre* | PMID: 7624356 | En1tm2(cre)Wrst | |
| Genetic reagent (M. musculus) | *En1-Cre* | JAX stock #007916 | En1tm2(cre)Wrst | |
| Genetic reagent (M. musculus) | R26RLacZ | PMID: 25594525 | | |
| Genetic reagent (M. musculus) | *Srd5a3Floxed* | IMPC - ES cell clone EPD0575_2_F01 carrying the Srd5a3tm1a (EUCOMM)Wtsi allele | Srd5a3tm1a (EUCOMM)Wtsi | |
| Antibody | BiP | Enzo | ADI-SPA-826 | 1:50 |
| Antibody | Calbindin | Swant | CB-38A | 1:10.000 |
| Antibody | GFAP | Millipore | AB5804; RRID:AB_2109645 | 1:1.000 |
| Antibody | L1CAM | Abcam | AB24345; RRID:AB_448025 | 1:2.000 |
| Antibody | L1CAM | *Appel et al. (1995)* | 557.B6 | Membrane staining |
| Antibody | Laminin | Sigma | L9393; RRID:AB_477163 | 1:33 |
| Antibody | LAMP1 | Abcam | AB24170; RRID:AB_775978 | 1:2.000 |
| Antibody | NeuN | Millipore | MAB377; RRID:AB_2298772 | 1:200 |
| Antibody | NrCAM | Abcam | AB24344; RRID:AB_448024 | 1:2.000/Also used for membrane staining |
| Antibody | Tuj1 | Sigma | T2200 | 1:2.000 |
| Antibody | β-actin | Thermo Fisher | AM4302; RRID:AB_2536382 | 1:20.000 |
| Chemical compound, drug | D-Lysine | Sigma | P6407 | |
| Chemical compound, drug | Fibronectin | Sigma | F2006 | |
| Chemical compound, drug | Laminin | Sigma | 11243217001 | Coating reagent |
| Chemical compound, drug | Lectin ConA | Sigma | C2272 | 1:2.000 |

*Continued on next page*

*Continued*

| Reagent type (species) or resource | Designation | Source or reference | Identifiers | Additional information |
|---|---|---|---|---|
| Chemical compound, drug | Lectin ConA | Sigma | C2010 | Glycoproteomics reagent |
| Chemical compound, drug | Lectin RCA120 | CliniSciences | L-1080 | Glycoproteomics reagent |
| Chemical compound, drug | Lectin SNA | Vector laboratories | B-1305 | 1:2.000 |
| Chemical compound, drug | Lectin WGA | Sigma | L9640 | Glycoproteomics reagent |
| Chemical compound, drug | PNGase | New England BioLabs | P0704L | |
| Chemical compound, drug | Tamoxifen | Sigma | H6278 | 1 μM |
| Chemical compound, drug | Tunicamycin | BioTechne | 3516/10 | 0.01 μg/mL |
| Chemical compound, drug | WATER-18O, 97 ATOM % 18O | Sigma | 329878 | Glycoproteomics reagent |
| Commercial assay or kit | GeneChip Mouse Transcriptome Array 1.0 | Affimetrix | MTA1.0 | |
| Peptide, recombinant protein | L1CAM-Fc | R and D | 777-NC | |
| Peptide, recombinant protein | NrCAM-Fc | R and D | 2034-NR | |
| Software, algorithm | ParTek Genomics Suite | ParTek | | |
| Other | IncuCyte | Essen Bioscience | IncuCyte S2 with NeuroTrack module | |

## Animal care

Animals were used in compliance with the French Animal Care and Use Committee from the Paris Descartes University (APAFIS#961 – 201506231137361). All behavioural analysis were performed in compliance with the guidelines of the French Ministry of Agriculture for experiments with laboratory animals. The experimental protocol and euthanasia have been approved by the Ethical Committee 27, registered at the French ministry of research.

## Generation of the *Srd5a3* conditional KO mouse

Embryonic stem (ES) cell clone EPD0575_2_F01 carrying the *Srd5a3*tm1a(EUCOMM)Wtsi allele was acquired from the International Mouse Phenotyping Consortium (IMPC; http://www.mousepheno-type.org/). After additional QC validation based on Southern-blot, ES cells were injected into C57BL/6 at the University of California, San Diego Transgenic and Gene targeting Core yielding chimeric mice. Chimera were bred to C57Bl/6 until germline transmission was successful to produce knockout first allele. Then, the *Srd5a3*tm1a mice were bred with ACTB:FLPe B6J (JAX stock #005703) Flp recombinase expressing transgenic mice to produce mice expressing a functional *Srd5a3* allele that retained flox sites surrounding exon 2 to 4 (out of the six exons) and including part of the enzymatic domain (PFAM, *Steroid_dh*). After outbreeding the Flp recombinase transgene, *Srd5a3* floxed mice were crossed with hemizygous CMV-Cre line transgenic mice (*Metzger et al., 1995*) to generate *Srd5a3* KO allele. *Srd5a3* floxed mice were bred with hemizygous *En1*tm2(cre)Wrst transgenic mice (JAX stock #007916) expressing Engrailed1 promoter driven Cre to produce cerebellum-specific deletion (En1Cre; *Srd5a3*fl/-) or with hemizygous Tg(Atoh1-cre)1Bfri (JAX stock #011104) expressing Atoh1 promoter driven Cre to produce cerebellum granule cell-specific deletion (Atoh1Cre; *Srd5a3*fl/-). Specific Cre expression was confirmed by breeding with a LacZ

reporter-carrier mouse line (R26R<sup>LacZ</sup> mouse). Efficient *Srd5a3* recombination was assessed by RT-PCR and RT-qPCR in the absence of any specific antibody.

## Mouse embryonic fibroblasts (MEFs) generation

For MEFs generation, *Srd5a3*<sup>fl/-</sup> mice were bred with CAGCre mice (B6.Cg-Tg(CAG-cre/Esr1*)5Amc/ J; CAGCre; *Srd5a3*<sup>fl/-</sup>). CAGCre;Srd5a3<sup>fl/-</sup> embryonic fibroblasts were isolated at E14.5 and immortalized by serial passaging as described by *Xu, 2005*. For *Srd5a3* recombination primary MEFs were treated for 4 days with tamoxifen (1 μM, H6278, Sigma) prior to immortalization. Efficient *Srd5a3* recombination was assessed by RT-qPCR after immortalization.

## Behavioral studies

For behavioral analysis, 15 *En1Cre; Srd5a3*<sup>fl/-</sup> and 15 control littermates, gender-matched, 2 – 3 months old were used. The number of animals per group was chosen as the optimal number likely required for conclusion of statistical significance, established from prior experience using the same behavioral tests (*Pereira et al., 2009*). Morris water maze (MWM) test was used to assess working memory. Prior to the test, the mouse swimming speed was analyzed. No differences in the swimming speed were detected. For MWM, mice were exposed twice to the same hidden platform for a total of eight sessions. The average latency for each mouse is represented. The improvement to find the platform on the second trial was evaluated. For motor coordination, classic accelerating rotarod testing was done for 10 min and for a total of three sessions, 24 hr apart. The first three falls from the rod during each session were annotated. A statistical Grubbs test was performed to verify the absence of significant outliers in each group.

## Histological analysis

Gross anatomical analyses and HE staining were performed as previously described (*Akizu et al., 2013*). The resulting slices were scanned with NanoZoomer-XR (Hamamatsu Photonics, Japan). We evaluated the cerebellum size difference by measuring the vermis area from mid-sagittal sections (*Pogoriler et al., 2006*). We identified the midline by the absence of fastigial nucleus and cerebellar peduncles. A single midline section was used to calculate the area with ImageJ. For quantification of the surface occupied by the clusters and their distribution, two mice and one slice per region were quantified per genotype. For IHC, the cerebellum was embedded in OCT. 12μm-thick slices were generated with cryostat. The following antibodies were used for IHC: NeuN (1:200, MAB377; Millipore), Calbindin (1:10.000; CB-38A, Swant), GFAP (1:1.000; AB5804, Millipore), laminin (1:33, L9393, Sigma) and BiP (1:50, ADI-SPA-826, Enzo). All samples were mounted with ProLong Gold Antifade Mountant with DAPI (Life Tech). Images were taken with confocal Leica SP8 STED and analyzed with ImageJ.

## Protein extraction and western blotting

P7 mouse cerebellum samples and MEFs were isolated in RIPA buffer (1% SDS; 0,1% for cell extracts) supplemented with EDTA-free protease inhibitor (11836170001, Sigma) and phosphatase inhibitor cocktail (4906845001, Sigma), homogenized by sonication (Bioruptor Pico sonication device – 8 cycles 30''ON/30''OFF) and centrifuged at 12.000 g at 4°C for 20 min. The supernatant was recovered and quantified with BCA (Life technologies). For cell extracts, RIPA was added directly to the flask and the cells were recovered with a cell scrapper followed by the same protocol. Equal amounts of protein were loaded from each sample in polyacrylamide gels. Gel transfer to nitrocellulose membranes was performed with the Trans-Blot Turbo Transfer System for 10 min at 1.3A and 25V. Membranes were blocked with 5% dry milk and incubated O/N at 4°C with the following antibodies: LAMP1 (1:2.000; AB24170, Abcam), L1CAM (1:2.000; AB24345, Abcam), NrCAM (1:2.000; AB24344, Abcam) and β-actin (1:20.000; AM4302, Thermo Fisher). Depending on the antibody suitability, the membrane was developed by HRP system (LAMP1, ChemiDoc XRS + System) or with fluorescent secondary antibodies (Odyssey CLx imaging system). All secondary antibodies were used at 1:10000. For far-western blotting, no blocking step was performed, and the membrane was directly incubated for 1 hr with biotinylated SNA (1:2.000; B-1305, Vector laboratories) or ConA (1:2000; C2272, Sigma) and posteriorly with IRDye 800CW-streptavidin for 1 extra hour (1:10.000;

926 – 32,230, LI-COR). All mouse WB results were replicated at least three times with several litters. Far-western blots were repeated at least twice with several litters.

## RNA extraction, RT-PCR, RT-qPCR and transcriptomic analysis

RNA was extracted with Trizol reagent (15,596–026, Thermo Fisher) according to manufacturer's instructions. For transcriptomic studies, *En1Cre;Srda5a3*$^{fl/-}$ P7 mice (n = 4) and control littermates (n = 4) were used. RNA quality was validated with Bioanalyzer 2100 (using Agilent RNA6000 nano chip kit) and 180 ng of total RNA were reverse transcribed using the GeneChip WT Plus Reagent Kit (Affymetrix). The resulting double strand cDNA was used for in vitro transcription with T7 RNA polymerase (WT cDNA synthesis and amplification kit, Affymetrix). After purification according to Affymetrix protocol, 5.5 ug of the cDNA obtained were fragmented and biotin-labelled using Terminal Transferase (WT terminal labelling kit, Affymetrix). cDNA was then hybridized to GeneChip Mouse Transcriptome Array 1.0 (MTA 1.0., Affymetrix) at 45°C for 17 hr. After O/N hybridization, chips were washed on the fluidic station FS450 following specific protocols (Affymetrix) and scanned using the GCS3000 7G. The scanned images were then analyzed with Expression Console software (Affymetrix) to obtain raw data (cel files) and metrics for Quality Controls. The observations of these metrics and the study of the distribution of raw data showed no outlier experiment. Robust multi-array average (RMA) normalization was obtained using R, and normalized data were subjected to statistical tests. For RT-PCR/RT-qPCR 1 µg of RNA was retrotranscribed into cDNA with SuperScript II reverse transcriptase (18064014, Thermo Fisher). qPCR was performed with PowerUp SYBR Green Master Mix (A25777, Thermo Fisher). The following primers were used: m$\beta$Actin (F 5'-TACAGC TTCACCACCACAGC-3'; R 5'-AAGGAAGGCTGGAAAAGAGC-3') and m*Srd5a3* (F 5'-CCGGGCTA TGGCTGGGTGG-3' and R 5'-CTGTCTCAGTGCCTCTAGGAATGG-3').

## Total proteomics and glycoproteomics

P7 *En1Cre;Srd5a3*$^{fl/-}$ (n = 4) and control (n = 4) littermates were used for total proteomics and the same number, but distinct mice, were used for glycoproteomics. The cerebellar protein extraction was performed as described above. For total proteomics, 100 µg of protein were processed by filter-aided sample preparation (FASP) protocol, as described previously (*Lipecka et al., 2016*). Briefly, samples were applied to 30 KDa MWCO centrifugal filter units (UFC503024, Amicon Ultra, Millipore) mixed with 200 uL of urea (UA) buffer (8M urea, 100 mM Tris-HCl pH 8.8) and centrifuged twice. The samples were incubated for 20 min in the dark with UA buffer containing 50 mM iodocetamide for alkylation. The filter units were subsequently washed twice with UA buffer and twice more with ABC buffer (50 mM ammonium bicarbonate). Peptide digestion was carried by incubation with trypsin (1:50) O/N at 37°C. The resulting peptides were collected by two washes with ABC buffer, vacuum dried and dissolved in 0.1% (v/v) trifluoroacetic acid with 10% acetonitrile.

Glycoproteome analysis was performed by FASP with an additional step of enrichment in N-glycopeptides by lectins, as described by M. Mann and colleagues (*Zielinska et al., 2010*). Briefly, 100 µg of trypsinized peptides were recovered in binding buffer (20 mM Tris/HCl pH 7.6, 1 mM MnCl$_2$, 1 mM CaCl$_2$, 0.5; NaCl) and incubated with a lectin mixture (90 µg ConA, 90 µg WGA and 71.5 µg RCA$_{120}$) for 1 hr. To elute the non-glycosylated peptides, not attached to the lectins, the filter units were washed four times with binding buffer and after with ABC solution in H$_2$O$_{18}$ (O188P, Eurositop). To release the N-glycopeptides from the lectins, the samples were incubated with PNGase diluted in H$_2$O$_{18}$ (P0704L, New England BioLabs) for 3 hr at 37°C. The N-glycopeptides were recovered by washing twice with ABC. All centrifugation steps were performed at 14.000 g at RT.

## LC-MS/MS analysis

For each run, estimated 0.5 µg were injected in a nanoRSLC-Q Exactive PLUS (Dionex RSLC Ultimate 3000, Thermo Scientific, Waltham, MA, USA). Peptides were separated on a 50 cm reversed-phase liquid chromatographic column (Pepmap C18, Dionex). Chromatography solvents were (A) 0.1% formic acid in water and (B) 80% acetonitrile, 0.08% formic acid. Peptides were eluted from the column with a linear gradient of 120 min from 5% A to 80% B followed by 27 min of column re-equilibration in 5% A. Two blanks, each with two 25 min-linear gradient, were run between samples to prevent carryover. Peptides eluting from the column were analyzed by data dependent MS/MS, using top-10 acquisition method. Briefly, the instrument settings were as follows: resolution was set to 70,000 for

MS scans and 17,500 for the data dependent MS/MS scans in order to increase speed. The MS AGC target was set to $3.10^6$ counts with 200 ms for the injection time, while MS/MS AGC target was set to $1.10^5$ with 120 ms for the injection time. The MS scan range was from 400 to 2000 m/z. Dynamic exclusion was set to 30 s. All analyses were performed in technical triplicate for each biological replicate.

## Total proteomics and glycoproteomics data analysis

The MS files were processed with MaxQuant software version 1.5.8.3 and searched with Andromeda search engine against the mouse subset from the UniProtKB/Swiss-Prot complete proteome database (release 2016_06). Statistical analysis and logo extractions were performed using Perseus version 1.5.5.3. Different thresholds were applied to total proteomics and glycoproteomics analysis given that the intensity of several peptides in total proteomics is used for determining protein intensity, while a single peptide in glycoproteomics is analysed at the time. For total proteomics, only proteins detected in all eight samples (4 controls and four mutants) were retained for statistical analysis, avoiding all data imputation. For comparative glycoproteomics, we retained glycosites detected in at least 3 out of 4 control samples. Additionally, we selected proteins that were specifically detected in control or mutant samples by retaining proteins detected solely in at least 3 samples of one group. Both FDR and p-value (q-value <0,05, paired student t-test) was used for total proteomics, whereas the p-value (<0,05, unpaired student t-test) was used for N-glycopeptides. As a database for N-glycoproteins and number of N-glycosylation sites per protein (qualitative dataset, reference glycoproteomic dataset), the data obtained by glycoproteomics was used: any glycopeptide detected in at least two control samples was considered as potentially N-glycosylated (*Supplementary File 2*). Sites that were not previously described in Zielinska et al. or not predicted by uniprot were classified as likely novel N-glycosylation sites. Volcano plots were generated using the VolcanoShiny app (https://hardingnj.shinyapps.io/volcanoshiny/). PCA and variance analysis were done with the Partek Genomics Suite software. For homologous IgSF-CAM proteins (CNTNs, L1CAMs and NCAMs), HomoloGene and the study from Chen et al. (*Chen and Zhou, 2010*) were used.

## Granule cell culture

Cerebellar granule cells were isolated from P7 cerebellum following the Manzini and colleagues protocol(*Lee et al., 2009*). Cells were kept at 37°C in 24 or 48-well plates for at least 24 hr. To detect expression of proteins at the plasma membrane, the protocol published by Carrodus et al. was followed(*Carrodus et al., 2014*). The rat monoclonal anti-L1CAM antibody (557.B6; 1:100) has been described (*Appel et al., 1995*). The anti-NrCAM antibody was purchased from Abcam (AB24344, 1:400). For binding with L1CAM-Fc (10 µg/mL; 777-NC, R and D) and NrCAM-Fc chimeras (10 µg/mL; 2034-NR, R and D), the cells were incubated at 37°C with the chimeras for 4 hr and 10 min, respectively, in granule cell culture media. Both the cell surface protein expression and binding assays were repeated three times with biological replicates. For each experiment, at least 20 neurons per mice were analyzed. The differences measured between control and mutant cells and expressed as a percentage in the main text represent the average of all the repeated experiments. Quantification and normalization were performed using imageJ and Tuj1 staining. For tunicamycin treatment, the cells were incubated for 16 hr with 0.01 µg /mL tunicamycin in culture media prior ICC. For surface coating, 48 or 24-well plates were incubated at 4°C overnight with the coating solution, followed by 3 PBS washes, blocking for 30 min with BSA to inhibit non-specific binding (10 mg/mL) and three more PBS washes. The coatings used were: human recombinant L1CAM (10 µg/mL; 777-NC, R and D), human recombinant NrCAM (5 µg/mL; 2034-NR, R and D), fibronectin (50 µg/mL; F2006, Sigma) and Laminin and poly-D-Lysine (2 µg/mL and 30 µg/mL; 11243217001 and P6407, respectively, Sigma). Cell and neurite dynamics were measured every 3 hr by live cell imaging (Incucyte ZOOM with Neurotracker module, Essen Bioscience). Neurite branching points and neurite length were normalized to the total number of cell body clusters. As long as GCs do not arborize, the neurite branching points parameter provided by the Incucyte software was translated as neurite number. For GCs re-aggregates, cells were isolated and plated for 24 hr in un-coated surfaces ($2 \times 10^6$ cells/cm$^2$) to promote aggregation. The aggregates were then collected and seeded on the coated surface for 24 hr. Sholl analysis (using the ImageJ plugin) was performed to quantify the

aggregates' neurite number as the number of neurites crossing each circle. For migration analyses, the cells within the same circles used in Sholl analyses but the first circle, were counted (*Tanaka et al., 2004*). Five different aggregates were quantified per mouse and coating condition.

## Human-induced pluripotent stem cells (hiPSCs) *SRD5A3*$^{-/-}$ generation and culture

iPSCs were derived from hPBMCs from a control male donor via Cyto-Tune Sendai virus reprogramming. Cells were cultured at 37°C on vitronectin-coated (10 μg/mL; 07180, Stem Cell) dishes with mTeSR media (Stem Cell). *SRD5A3* KO hiPSCs clones were generated by CRISPR/Cas9. sgRNA (inserted into a GFP-containing PX458 plasmid, Addgene) targeting the first exon of the gene were generated via the CRISPOR website and validated in T293 HEK cells by Sanger sequencing combined with tides analysis (https://tide-calculator.nki.nl/; data not shown). hiPSCs were transfected by nucleofection (Amaxa 4D, Lonza) and transfected cells (GFP+) were isolated by FACs (BD FACSAria II SORP, BD Biosciences) to generate single-cell-of-origin colonies. DNA was extracted from a piece of each colony by ZR-Duet DNA MiniPrep (D7003, Zymo) and sequenced. After selection, SOX2 (1:2000; AB5603, Millipore) and OCT4 (1:100; SC5279, Santa Cruz) immunostaining was used to confirm pluripotency (*Figure 5—figure supplement 2*). No major chromosomal abnormalities were detected by CGH array (60K, data not shown).

## hiPSCs differentiation towards late cortical progenitors (LCPs)

Differentiation of hiPSCs towards LCPs and cortical projection neurons (CPNs) was carried as described by Benchoua and colleagues (*Boissart et al., 2013*). Briefly, iPSCs colonies were transferred to a non-coated dish with neural induction media (N2B27 with FGF2 and double SMAD inhibition by SB431542 and LDN193189, Stem Cell) for 6 hr and were afterwards transferred onto polyornithine and laminincoated dishes. Following neural rosette formation (12 – 15 days), the cells were passaged onto a geltrex-coated (A1413301, Life Tech) flask (LCPs P1) and were further cultured with N2B27 supplemented with FGF2, EGF and BDNF (Stem Cell). SOX2 and Nestin (1:250; N5413, Sigma) staining confirmed LCPs multipotency. Neuroectodermal origin of the emerging neural progenitor-like cells was assessed by HNK1/P75 FACs staining (data not shown). LCPs were further differentiated towards CPNs by growth factor withdrawal for 15 days. Neuronal identity was assessed with Tuj1 (1:2000; T2200, Sigma) staining (data not shown). WB results were replicated with two different clones of control and *SRD5A3* mutant hiPSCs.

## Electronic microscopy experiments

P21 *En1Cre; Srd5a3*$^{fl/-}$ (n = 1), *Atoh1Cre; Srd5a3*$^{fl/-}$ (n = 2) and control littermates (n = 3) were perfused with 4% PFA/2% glutaraldehyde. The cerebellum was kept in the same solution for at least one week. Sagittal slices, less than 1 mm thick, were post-fixed with 1% osmium tetroxide in 0.1 M phosphate buffer and then dehydrated in ethanol. After 10 min in a 1:2 mixture of epoxy propane and epoxy resin and 10 min in epon, samples were covered by upside down gelatin capsules filled with freshly prepared epoxy resin and polymerized at 60°C for 24 hr. After heat separation, ultrathin sections of 90 nm were cut with an ultra-microtome (Reichert ultracut S), stained with uranyl acetate and Reynold's lead and observed with a transmission electron microscope (JEOL 1011). Acquisition was performed with a Gatan Orius 1000 CCD camera.

## Statistics

For mouse experiments, no statistical methods were used to predetermine sample size. No animals or samples were excluded from the analysis. All mouse experiments included at least two different litters. For mouse behavioral analysis, one way ANOVA was used. Regular enrichment analyses were performed by Fisher exact test (significance 0.05). For multiplicity and NxS motif correlation, the Pearson coefficient was used. For enrichment in highly N-glycosylated proteins, two-tailed Mann-Whitney test was performed. Statistical differences between groups were assessed by two-tailed student t-test. For all statistical tests, confidence intervals were set at 95%. All results are presented in mean value and standard deviation was used to calculate the error bars.

## Data availability

Full transcriptomic data is publicly available at ArrayExpress (accession no. E-MTAB-6861). Total proteomics and glycoproteomics data are available via ProteomeXchange with identifier PXD009906.

# Acknowledgements

We thank JG Gleeson for the support to generate *Srd5a3* conditional mouse and helpful advices. We are grateful to L Goutebroze, F Francis, N Altin, K Siquier, K Radjamani for valuable discussions or technical help. The project is funded by the French National Research Agency ANR-16-CE12-0005-01 (VC), la Fondation pour la Recherche Médicale FRM-DEQ20160334938 (LC), the association Connaître les Syndromes Cérébelleux CSC (VC), the program ERA-NET 643578-EURO-CDG2 and E-Rare-3 (FF and CT), Paris Descartes University BioSPC program (DM-C), Imagine Foundation (DM-C), Imagine International PhD program with Bettencourt-Schueller-foundation (EU). We thank Cochin genomic (GENOM'IC), electron microscopy core facilities and the Imagine animal facilities (LEAT) for services and support. We also thank the Animal Histology and Morphology Core Facility of SFR Necker (Inserm US24). We also thank L. Legeai-Mallet and C Colnot for providing the CMV-Cre and the Rosa26-LacZ mouse, respectively and Nicolas Cagnard for statistical advice.

# Additional information

## Funding

| Funder | Grant reference number | Author |
| --- | --- | --- |
| Université Paris Descartes | BioSPC program | Daniel Medina-Cano |
| European Commission | ERA-NET 643578-EURO-CDG2 | Christian Thiel François Foulquier |
| European Commission | ERA-NET E-Rare-3 | Christian Thiel François Foulquier |
| Fondation pour la Recherche Médicale | FRM-DEQ20160334938 | Laurence Colleaux |
| Agence Nationale de la Recherche | ANR-16-CE12- 0005-01 | Vincent Cantagrel |
| Association Connaître les Syndromes Cérébelleux | | Vincent Cantagrel |

The funders had no role in study design, data collection and interpretation, or the decision to submit the work for publication.

## Author contributions

Daniel Medina-Cano, Data curation, Formal analysis, Validation, Investigation, Visualization, Methodology, Writing—original draft; Ekin Ucuncu, Data curation, Formal analysis, Methodology; Lam Son Nguyen, Joanna Lipecka, Christian Thiel, Formal analysis, Methodology; Michael Nicouleau, Data curation, Formal analysis; Jean-Charles Bizot, Data curation, Formal analysis, Supervision; François Foulquier, Data curation, Investigation; Nathalie Lefort, Formal analysis; Catherine Faivre-Sarrailh, Conceptualization, Resources, Supervision, Methodology, Writing—review and editing; Laurence Colleaux, Formal analysis, Funding acquisition, Project administration, Writing—review and editing; Ida Chiara Guerrera, Resources, Data curation, Formal analysis, Supervision, Visualization, Methodology; Vincent Cantagrel, Conceptualization, Formal analysis, Supervision, Funding acquisition, Validation, Investigation, Methodology, Writing—original draft, Writing—review and editing

## Author ORCIDs

Vincent Cantagrel [ID] http://orcid.org/0000-0002-5180-4848

## Ethics

Human subjects: Primary human mononuclear cells were isolated from peripheral blood and were obtained using Institutional Review Board-approved consent forms. Induced human pluripotent stem

cells protocols, samples collection and storage were approved by the French research ministry, identification DC 2015-2595 obtained on Sept-05 2016.

Animal experimentation: In this study, animals were used in compliance with the French Animal Care and Use Committee from the Paris Descartes University (APAFIS#961-201506231137361). All behavioural analysis were performed in compliance with the guidelines of the French Ministry of Agriculture for experiments with laboratory animals. The experimental protocol and euthanasia have been approved by the Ethical Committee 27, registered at the French ministry of research.

### Decision letter and Author response
Decision letter https://doi.org/10.7554/eLife.38309.026
Author response https://doi.org/10.7554/eLife.38309.027

## Additional files

### Supplementary files
• Supplementary file 1. Total proteomics. Normalized and log2 transformed intensities of the 1982 proteins detected by total proteomics and their relative abundance levels in the En1-Cre; *Srd5a3*^fl/- mutant cerebellum.
DOI: https://doi.org/10.7554/eLife.38309.017

• Supplementary file 2. Glycoproteomics. All of the N-glycosylation sites detected in at least 2 out of 4 control samples are included and classified based on their presence in PMID: 20510933 or if they are predicted by uniprot (reference glycoproteomic dataset, spread sheet 1). Sequon analyses on the 1665 N-glycopeptides within the glycoproteomic dataset (spread sheet 2). Gene ontology analyses on the N-glycosylation sites within the reference glycoproteomic dataset predicted by uniprot and/or found in PMID: 20510933 and on the N-glycosylation sites not present in none of them (novel sites, spread sheet 3). 1404 N-glycopeptides detected in at least 3 out of 4 control samples and their corresponding abundance levels in the mutant samples; the intensity for each N-glycopeptide are log2 transformed (comparative glycoproteomics, spread sheet 4). Sequon analyses on the comparative glycoproteomics dataset (spread sheet 5).
DOI: https://doi.org/10.7554/eLife.38309.018

• Supplementary file 3. Total proteomics and glycoproteomics. List, normalized intensities and relative abundance of the 24 peptides detected in total proteomics and with decreased abundance in glycoproteomics in the mutant.
DOI: https://doi.org/10.7554/eLife.38309.019

• Transparent reporting form
DOI: https://doi.org/10.7554/eLife.38309.020

### Data availability
Expression microarray data are publicly available at ArrayExpress accession number E-MTAB-6861. Proteomics and Glycoproteomics data are included within supplementary files 1, 2 and 3 and full proteomics and glycoproteomics data are available via ProteomeXchange with identifier PXD009906.

The following datasets were generated:

| Author(s) | Year | Dataset title | Dataset URL | Database, license, and accessibility information |
|---|---|---|---|---|
| Medina-Cano D, Lipecka J, Guerrera IC, Cantagrel V | 2018 | GlycoProteomics in CDG (congenital disorder of glycosylation) | https://www.ebi.ac.uk/pride/archive/projects/PXD009906 | Publicly available at EBI PRIDE (accession no. PXD009906) |
| Medina-Cano D., Cantagrel V. | 2018 | Transcriptomic analysis of the developing cerebellum in a mouse model for SRD5A3-CDG | https://www.ebi.ac.uk/arrayexpress/experiments/E-MTAB-6861/ | Publicly available at EMBL-EBI Array Express (accession no. E-MTAB-6861) |

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
