## [Decision Letter]

Thank you for submitting your article "High N-glycan multiplicity is critical for neuronal adhesion and sensitizes the cerebellum to N-glycosylation defect" for consideration by *eLife*. Your article has been reviewed by two peer reviewers, and the evaluation has been overseen by a Reviewing Editor and Jonathan Cooper as the Senior Editor. The following individual involved in review of your submission has agreed to reveal his identity: Kevin M Wright (Reviewer #2).

The Reviewing Editor has drafted this decision to help you prepare a revised submission.

Summary:

In the manuscript by Medina-Cano, the authors explore the neurodevelopmental requirement for *Srd5a3*, an enzyme required for N-glycosylation and associated with congenital disorders of glycosylation (CDG). They show that mice lacking *Srd5a3* in the cerebellum develop ataxia and cerebellar developmental phenotypes consistent with human patients with *Srd5a3*-associated CDG. Interestingly, they find that the overall levels of glycosylation are not largely altered, but that loss of *Srd5a3* seems to preferentially affect the glycosylation of proteins with high multiplicity of glycosylation sites. They go on to show that glycosylation of L1CAM and NrCAM, two members of the IgSF family of cell adhesion molecules, is reduced in *Srd5a3* cKOs. This results in reduced adhesion of *Srd5a3* cKO neurons on IgSF substrates in vitro, which likely contributes to the neuronal migration and axonal defects seen in the *Srda3* cKO cerebellum.

Two major observations are:

1) Proteins with higher number of N-glycosylation sites have their glycosylation most reduced, amongst these being highly glycosylated immunoglobulin family CAMS (IgSF-CAMs) and

2) The neural migration defect of cerebellar granule neurons is likely to be due to the reduced glycosylation of IgSF-CAMs.

Overall, this study sheds light on the mechanism underlying a form of CDG that is not well understood. The data are mostly clear and provide interesting and unexpected results related to specific requirement for *Srd5a3* in high multiplicity glycosylation events. While L1 and NrCAM (and other IgSF family members) are certainly not the only proteins affected by the loss of *Srd5a3*, their hypoglycosylation does provide a reasonable explanation for the phenotypes observed in the cKO mice. There are some points that should be addressed to strengthen the arguments in the paper.

Essential revisions:

1) The experiments demonstrating a function role of the IgSF-CAMs are not definitive. In the animals, the authors identify a migration defect from the EGL to the IGL, with some changes in parallel fiber growth. They culture granule cell clusters, and measure neurite sprouting as "normalized neurite number". While they do show a reduction in this parameter when cells are plated on IgSF-CAMs, it is not clear how the parameter of neurite number relates to either neural migration or the altered trajectories of parallel fibers. There are other methods to measure GC migration in culture (e.g., from the Hatten lab), and also effects on neurite outgrowth, and it would seem these would be more appropriate.

2) The authors conclude that the biological actions (migration, axonal pathfinding) are due to hypoglycosylation of IgSF-CAMs that reduces interactions. Because there are lower levels of some affected proteins, it is important to show that there is no reduction of surface expression of these IgSF-CAMS. In addition, it should be possible to use a more direct adhesion assay to investigate the interaction of the hypoglycosylated protein with the WT, both using cells and covaspheres as was done many years ago in the Edelman lab for NrCAM. It is stated in the Discussion (third paragraph) that levels of full length L1CAM are not changed, however reduced glycosylation can affect processing, trafficking, and/or function. It should be straightforward for the authors to examine trafficking/cell-surface expression of endogenous L1CAM in CGNs by either cell-surface biotinylation or immunocytochemistry on non-permeabilized neurons.

3) It is not clear how many times certain experiments were repeated or whether they used power calculations to estimate sample size.

---

## [Author Response]

Essential revisions:1) The experiments demonstrating a function role of the IgSF-CAMs are not definitive. In the animals, the authors identify a migration defect from the EGL to the IGL, with some changes in parallel fiber growth. They culture granule cell clusters, and measure neurite sprouting as "normalized neurite number". While they do show a reduction in this parameter when cells are plated on IgSF-CAMs, it is not clear how the parameter of neurite number relates to either neural migration or the altered trajectories of parallel fibers. There are other methods to measure GC migration in culture (e.g., from the Hatten lab), and also effects on neurite outgrowth, and it would seem these would be more appropriate.

In the experiment that is mentioned, we studied neurite outgrowth as a readout for IgSF-CAM mediated cell adhesion as done in the literature (Stoeckli et al., 2013). In response to this comment we provide additional data from previous work and from a new experiment to measure neurite outgrowth and neural migration as suggested. The live cell imaging system that we used allows to compute automatically, in a high-throughput, several parameters including neurite number and length (see examples: Park et al., 2017; Tortoriello et al., 2014). In addition to the neurite number defect mentioned above, we also identified a significant defect in neurite length in *Srd5a3* mutant although not systematically more severe on IgSF-CAM-specific versus non-specific coating. This information is now included in Figure 7—figure supplement 1B-D and suggests that the neurite initiation process is more specifically affected than neurite extension which is compatible with the known role of L1CAM in neuritogenesis (Nishimura et al., 2003). To provide an alternative quantification for this phenotype, we used granule cell re-aggregates, a method frequently used to analyze cell migration under different coating conditions (Buttiglione et al., 1998; Katic et al., 2014; Kholmanskikh et al., 2003). We used this technique with L1CAM versus Laminin/D-Lysine coating and automatically counted the number of migrating cells and neurites at different distances from the cell re-aggregate. Strikingly, the absence of *Srd5a3* almost abrogates the ability of aggregated neurons to develop neurites on L1CAM coating. By contrast, the neurite defect is only close to significance on Laminin coating. These results are now added in the new Figure 7D, E and illustrate further the specificity of the IgSF-CAM cell adhesion defect in neurite outgrowth. This figure replaces the previous Figure 6 where the panel 6A was removed as redundant and less informative compared to the new Figure 7D, E. It is known that postmitotic granule cells will first rely on CNTN2 to develop their parallel fibers before starting their migration from the external granule cell layer (EGL) to the internal one. Interestingly, CNTN2 is an IgSF-CAM member that interacts both with L1CAM and NrCAM (Pollerberg et al., 2013) and its absence is associated with granule cells ectopias very similar to the ones observed in *Srd5a3* cKO cerebellum (Xenaki et al., 2011). This observation suggests that a defect in the development of neurites/axons could prevent these cells from proceeding with their final migration.

2) The authors conclude that the biological actions (migration, axonal pathfinding) are due to hypoglycosylation of IgSF-CAMs that reduces interactions. Because there are lower levels of some affected proteins, it is important to show that there is no reduction of surface expression of these IgSF-CAMS. In addition, it should be possible to use a more direct adhesion assay to investigate the interaction of the hypoglycosylated protein with the WT, both using cells and covaspheres as was done many years ago in the Edelman lab for NrCAM. It is stated in the Discussion (third paragraph) that levels of full length L1CAM are not changed, however reduced glycosylation can affect processing, trafficking, and/or function. It should be straightforward for the authors to examine trafficking/cell-surface expression of endogenous L1CAM in CGNs by either cell-surface biotinylation or immunocytochemistry on non-permeabilized neurons.

To address this important question related to the consequences of the observed hypoglycosylation, we studied the cell surface expression of L1CAM and NrCAM in nonpermeabilized cells using antibodies directed against the extracellular domain (see Materials and methods). We quantified the expression of these IgSF-CAMs that are well detected along the axons and at the growth cone but with 30% and 39% decrease for L1CAM and NrCAM, respectively (average of four different experiments) in *Srd5a3* cKO neurons. We found that a similar effect can be recapitulated using Tunicamycin suggesting that the disruption of other steps of the N-glycosylation pathway as well as specific drugs can efficiently alter the expression of these glycoproteins.

In order to investigate how the glycosylation impairment impacts global IgSF-CAM adhesion property at the cell level, we used chimeras made up of the extra-cellular part of L1CAM and NrCAM fused with Fc domain of IgG that were incubated with cultured granule cells. This assay is based on the same principle as the covasphere/coated bead assay that was suggested, but the detection of the bound IgSF-CAM is performed using the classical cell binding assay with Fc chimeras and anti-Fc antibody (Castellani et al., 2002; Faivre-Sarrailh et al., 1999). Specific binding of the IgSF-CAM chimeras was first validated in HEK cells transfected with the corresponding IgSF-CAM (not shown). The binding was then evaluated in primary GCs and a 38% and 56% reduction in binding of L1CAM and NrCAM chimeras, respectively (average of three independent experiments), was observed in mutant cells. The previously identified decrease in membrane expression is likely to have a major influence on this parameter. Altogether, these results suggest that the global outcome of the hypoglycosylation of these IgSF-CAMs is a decrease in membrane expression associated with a decrease in cell adhesion property and these data are now presented in an additional figure, the new Figure 6.

Importantly, these results reflect the behavior of a mixture of various glycoforms of each IgSF-CAM (visible on western-blot or in our proteomic data). They are consistent with previous studies of individual glycosylation sites in IgSF-CAM members where their mutagenesis is more often associated with decrease in membrane expression than severe loss of folding/stability (Bonnon et al., 2003; Bonnon et al., 2005; Jiménez et al., 2005) or direct effect on adhesion property (Fogel et al., 2010) although this last point has been much less investigated.

3) It is not clear how many times certain experiments were repeated or whether they used power calculations to estimate sample size.

We further precise the number of times each experiment was repeated in the corresponding Materials and methods section or figure legend. Each replication was successful. For the mouse behavior test no power calculation was used but the number of animals per group was chosen as the optimal number likely required for conclusion of statistical significance, established from prior experience using the same behavioral tests (Pereira et al., 2009).

References:

Bonnon C, Bel C, goutebroze L, Maigret B, Girault JA, Faivre-Sarrailh C. (2007)

PGY Repeats and N-Glycans Govern the Trafficking of Paranodin and Its Selective Association with Contactin and Neurofascin-155. Molecular Biology of the Cell 18(1), doi: 10.1091/mbc.e06-06-0570

Bonnon C, Goutebroze L, Denisenko-Nehrbass N, Girault JA, Faivre-Sarrailh C. (2003) The Paranodal Complex of F3/Contactin and Caspr/Paranodin Traffics to the Cell Surface via a Non-conventional Pathway. The Journal of Biological Chemistry, 278, 48339-48347. doi: 10.1074/jbc.M309120200

Buttiglione M, Revest JM, Pavlou O, Karagogeos D, Furley A, Rougon G, Faivre-Sarrailh C. (1998) A Functional Interaction between the Neuronal Adhesion Molecules TAG-1 and F3 Modulates Neurite Outgrowth and Fasciculation of Cerebellar Granule Cells. Journal of Neuroscience 1 September 1998, 18 (17) 6853-6870; DOI: 10.1523/JNEUROSCI.18-17-06853.1998

Castellani V., De Angelis E., Kenwrick S., Rougon G. (2002) Cis and trans interactions of L1 with neuropilin-1 control axonal responses to semaphorin 3A. The EMBO Journal (2002) 21, 6348-6357, DOI 10.1093/emboj/cdf645

Faivre-Sarrailh C., Falk J., Pollerberg E., Schachner M., Ron G. (1999) NrCAM, cerebellar granule cell receptor for the neuronal adhesion molecule F3, displays an actin-dependent mobility in growth cones. J Cell Sci 1999 112: 3015-3027

Fogel AI, Li Y, Giza J, Wang Q, Lak TT, Modis Y, Biedere T. (2010) N-Glycosylation at the SynCAM (Synaptic Cell Adhesion Molecule) Immunoglobulin Interface Modulates Synaptic Adhesion. The Journal of Biological Chemistry, 285 34864-34874. doi: 10.1074/jbc.M110.120865

Jiménez D, Roda-Navarro P, Springer TA, Casasnovas JM. (2005) Contribution of N-Linked Glycans to the Conformation and Function of Intercellular Adhesion Molecules (ICAMs). The Journal of Biological Chemistry 280, 5854-5861. doi: 10.1074/jbc.M412104200

Kholmanskikh SS, Dobrin JS, Wynshaw-Boris A, Letourneau PC, Ross ME. (2003) Disregulated RhoGTPases and Actin Cytoskeleton Contribute to the Migration Defect in Lis1-Deficient Neurons. Journal of Neuroscience 24 September 2003, 23 (25) 8673-8681; DOI: 10.1523/JNEUROSCI.23-25-08673.2003

Nishimura K, Yoshihara F, Tojima T, Ooashi N, Yoon W, Mikoshiba K, Bennett V, Kamiguchi H. (2003) L1-dependent neuritogenesis involves ankyrinB that mediates L1-CAM coupling with retrograde actin flow. J. Cell Biology. The Journal of Cell Biology Dec 2003, 163 (5) 1077-1088; DOI: 10.1083/jcb.200303060

Park, N. I., Guilhamon, P., Desai, K., McAdam, R. F., Langille, E., O'Connor, M., Lan, X., Whetstone, H., Coutinho, F. J., Vanner, R. J. et al. (2017). ASCL1 reorganizes chromatin to direct neuronal fate and suppress tumorigenicity of glioblastoma stem cells. Cell Stem Cell 21, 411. doi:10.1016/j.stem.2017.08.008

Stoeckli ET, Kilinc D, Kunz B, Kunz S, Lee GU, Martines E, et al. (2013) Analysis of cell-cell contact mediated by Ig superfamily cell adhesion molecules. Curr Protoc Cell Biol. 2013;61:9.5.1–9.5.85.

Tortoriello G, Morris CV, Alpar A, Fuzik J, Shirran SL, Calvigioni D, Keimpema E, Botting CH, Reinecke K, Herdegen T, Courtney M, Hurd YL, Harkany T. (2014) Miswiring the brain: Δ9-tetrahydrocannabinol disrupts cortical development by inducing an SCG10/stathmin-2 degradation pathway. EMBO J.2014;33(7):668–685